# A normative theory of social conflict

**Sergey Shuvaev**[1], **Evgeny Amelchenko**[2], **Dmitry Smagin**[3],
**Natalia Kudryavtseva**[3], **Grigori Enikolopov**[2], and **Alexei Koulakov**[1]

[1]Cold Spring Harbor Laboratory, Cold Spring Harbor, NY, USA; [2]Department of Anesthesiology and Center for Developmental Genetics, Stony Brook University, Stony Brook, NY, USA; [3]Institute of Cytology and Genetics, Siberian Branch of Russian Academy of Sciences, Novosibirsk, Russia.
`koulakov@cshl.edu`

## Abstract

Social conflict is a survival mechanism yielding both normal and pathological behaviors. To understand its underlying principles, we collected behavioral and whole-brain neural data from mice advancing through stages of social conflict. We modeled the animals' interactions as a normal-form game using Bayesian inference to account for the partial observability of animals' strengths. We find that our behavioral and neural data are consistent with the first-level Theory of Mind (1-ToM) model where mice form "primary" beliefs about the strengths of all mice involved and "secondary" beliefs that estimate the beliefs of their opponents. Our model identifies the brain regions that carry the information about these beliefs and offers a framework for studies of social behaviors in partially observable settings.

## 1 Introduction

Research into conflict behavior is an established domain within cognitive science and neuroscience with a focus on exploring the origins of pathological aggression and defeat (Archer, 1988; Rosell and Siever, 2015). The broad interest in this topic is driven by the pervasive occurrence of pathologically aggressive behaviors such as school bullying and prison hostility that remain a significant societal concern. Gaining insight into the neural and behavioral foundations of conflict behavior is a crucial step toward mitigating the adverse consequences of aggression and preventing instances of hostility.

Toward this goal, the field of conflict studies has adopted an experimental approach. Researchers have monitored rodent behavior and recorded corresponding brain activity, as rodents naturally exhibit aggressive behaviors allowing experimenters to minimize additional stress for animals during data collection. This approach has yielded valuable insights into the factors that promote aggression (Archer, 1988; Wang and Anderson, 2010; Lorenz, 2005) and led to identifying brain regions affected by conflict (Aleyasin et al., 2018; Diaz and Lin, 2020; Wei et al., 2021). The studies in this field, however, rarely focused on the dynamic nature of aggressiveness in individuals, thus constraining our ability to understand and predict harmful behaviors. Here, we use behavioral data modeling to explore the strategies of aggressive behavior in individual mice over prolonged periods of time.

Our study contributes to the field as follows. We collected large-scale data on behavior and whole-brain neural activity that follows the emergence of aggression and defeat in individual mice. We also introduced a family of models to characterize the learning rules, decision-making processes, and cost functions underlying aggression. These models were applied to the observed behaviors of individual mice to identify the best-fitting model. Using that model, we described the decision-making mechanism consistent with the emergence of aggression and submissiveness. We then conducted an analysis of the neural correlates of both explicit and hidden model variables across the entire brain, thus validating the hidden dynamics of the model and linking specific brain regions to individual model variables. While similar approaches were used in theoretical and computational neuroscience, they have not been previously applied in studies of conflict.

37th Conference on Neural Information Processing Systems (NeurIPS 2023).

Our findings show that behavioral and neural data describing chronic conflict aligns with the first-level Theory of Mind (1-ToM) model. According to this model, mice form "primary" beliefs about their own and their opponents' strengths along with "secondary" beliefs estimating their opponents' corresponding beliefs. These belief systems are updated with Bayes' rule based on the actions of opponents and the outcomes of aggressive encounters. We propose that, by representing Bayesian beliefs, animals in social conflict situations address the challenge of partial observability. Our model thus holds the potential for generalization to other social interactions characterized by uncertainty.

The proposed model not only provides insights into the development of pathological aggression and defeat but also offers a path toward their mitigation. The next steps will involve applying the model to newly available data and identifying the neural correlates of aggression-related variables within the brain. This will be followed by the exploration of connectivity changes between the identified brain regions, which may unveil the neural circuit governing decision-making in aggression. Ultimately, the shift from a normative to a circuit model of conflict may enable the design of policies geared toward reducing aggression in a variety of social settings.

## 2    Related work

**Behavioral biology of aggression.** Social conflict has been studied in humans, non-human primates, and mice – a model organism whose behavioral states can be manipulated. Mouse studies have investigated how social conflict leads to the formation, maintenance, and plasticity of aggressive and subordinate behavioral states in animals (Wong et al., 2016; Hashikawa et al., 2017). Such behavioral states can be modeled in the chronic social conflict paradigm where mice are allowed limited interactions over extended periods of time (Kudryavtseva, 2000; Miczek et al., 2001). Here, we use the chronic social conflict paradigm to induce various behavioral states in mice and monitor their conflict-related behaviors. As the previous studies of social conflict have shown the evolutionary preservation of its basic mechanisms (Wang and Anderson, 2010; Watanabe et al., 2017), we expect our results to be relevant to human behavior.

**Game theory.** Optimal behaviors of interacting agents are conventionally described in terms of game theory. Game theory considers rational agents developing their strategies to maximize rewards. The rewards received by the agents depend on their actions and the actions of their opponents. The acquisition of optimal strategies in games can be described by probabilities of available actions (Smith, 1982; Cressman et al., 2003). Such strategies of agents co-evolve to reinforce higher-reward actions until the rewards can't grow any further (Nash equilibrium). Game-theoretic approaches have been used in models of human and animal behaviors in multi-agent settings including agonistic interactions (Smith, 1974; Hofbauer et al., 1998; Wilson, 2000; Lorenz, 2005). Here, we use game theory to model agonistic interactions in the condition of chronic social conflict in mice.

**Beliefs and Theories of Mind.** Humans and animals in social settings have limited access to environmental variables, which can be modeled with the partially observable Markov decision process (POMDP) framework. To gain evidence about hidden variables, real-world agents may maintain probabilistic internal models of environment – the "beliefs" – based on which their actions can be considered rational, maximizing a reward function (Fahlman et al., 1983; Alefantis et al., 2021). The agents' rewards and beliefs can be inferred from their behavior using inverse control techniques that maximize the likelihood of the observed behavior based on a hidden dynamics model (Russell, 1998; Choi and Kim, 2011; Dvijotham and Todorov, 2010; Kwon et al., 2020). In biologically relevant multi-agent settings, beliefs are studied in the Theories of Mind (ToM) framework, proposing that humans and animals maintain beliefs about the beliefs of their adversaries (Baker et al., 2011) or aides (Khalvati et al., 2019). As previous studies were successful in inferring beliefs (Schmitt et al., 2017; Alefantis et al., 2021) and regressing them to neural activity in simulations (Wu et al., 2020) or low-resolution fMRI (Koster-Hale and Saxe, 2013), here we introduce a framework for the inference of beliefs in social setting and regress them to high-resolution whole-brain neural activity in mice.

**C-Fos as a whole-brain marker of neural activity.** The search for the brain regions accumulating evidence about the environment requires large-scale neural activity data. Such data can be obtained by monitoring the levels of c-Fos, an immediate early gene whose activation reflects neuronal activity (Sagar et al., 1988; Herrera and Robertson, 1996). C-Fos data lacks temporal resolution, yet it allows observing whole-brain activity at a high spatial resolution without using equipment that may affect animals' choices. Local expression of c-Fos has implicated several brain regions in agonistic

interactions (Hashikawa et al., 2017; Aleyasin et al., 2018; Diaz and Lin, 2020; Wei et al., 2021). Here, we use 3D light-sheet microscopy of the c-Fos signal in whole-brain samples (Renier et al., 2016) to identify the brain-wide neural activity in animals with varying exposure to social conflict. We compare our c-Fos data to the beliefs identified based on behavior in individual mice and report the regions that may be involved in the computation of conflict-related variables in the brain.

# 3 Methods: a normative POMDP framework for social behavior modeling

The goal of this work is to build a quantitative theory for the formation of social conflict-related behavioral states in mice. In Section 3.1 we introduce a mouse behavioral paradigm where we recorded the actions leading to different behavioral states and the brain activities corresponding to these states. In Sections 3.2 to 3.4 we define a normative POMDP-based framework for studying real-world social behaviors. In this framework, the game-theory optimal actions of agents rely on their beliefs, defined as distributional estimates of the hidden variables of the environment. In Section 4 we examine hypotheses about the reward schedule, information availability, and evidence accumulation related to social conflict. We compare our results to behavioral data in Section 4.1 and to neural data in Section 4.2. We discuss our findings in Appendix A. A detailed description of the methods is provided in Appendix B. Derivations and supplementary results are provided in Appendix C.

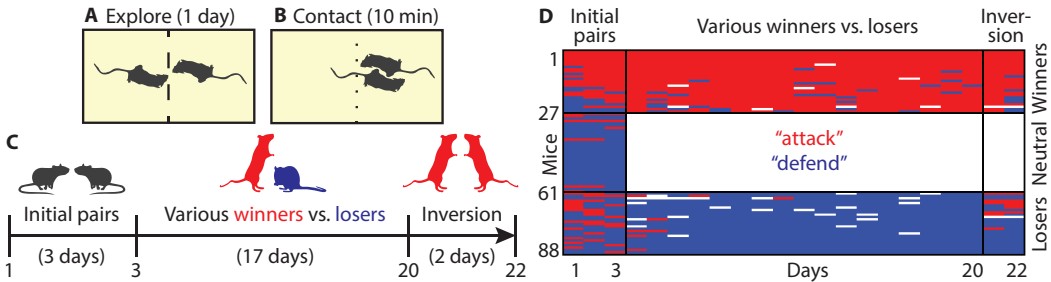

Figure 1: The chronic social conflict paradigm. (A-B) Stages of a single sensory contact event. (C) Stages of a multi-day experiment. (D) Recorded actions of mice in the experiment (white: no encounter scheduled).

## 3.1 Running example: the chronic social conflict paradigm

Our goal was to come up with a unified behavioral model explaining various stages of social conflict. As previous experiments have mostly focused on isolated aspects of social conflict (e.g. its pathological form only), the data needed for building a unified model was not readily available, necessitating the conduct of a new experiment. To this end, we implemented the chronic social conflict paradigm (Kudryavtseva, 2000; Kudryavtseva et al., 2014), a well-established experimental paradigm for studying aggression and conflict. Within the paradigm, we focused on conventionally studied stages of social conflict, i.e. the adaptive (3 days), intermediate (10 days), and pathological (20 days) aggression/defeat (Kudryavtseva et al., 2014). For completeness, we additionally considered the reversal of pathological behaviors (after 20 days) – the standard experimental perturbation of behavior. We used the conventional experimental parameters (Kudryavtseva et al., 2014).

We conducted our experiments as follows. Pairs of weight-matched (as a proxy for being strength-matched) mice were placed in cages separated by a perforated partition (Figure 1A). Once daily, the partition was removed for 10 minutes to enable agonistic interactions between mice (Figure 1B). After 3 days, mice were classified into "winning", "losing", and "neutral" (non-fighting) (Figure 1C,D). The participation of "neutral" mice in the experiment was discontinued. Afterward, each winning mouse remained in its cage, while each losing mouse was daily relocated to an unfamiliar cage with an unfamiliar winning mouse (Figure 1C). Regardless of no longer being weight-matched, mice have retained their winning or losing behavioral states, thus transitioning to the pathological regime of social conflict-related decision-making (Figure 1D). After 20 days of interactions, mice were exposed to opponents of equal behavioral state (Figure 1C). The newly formed pairs underwent two more days of agonistic interactions throughout which new win/lose relationships were established (Figure 1D). We provide more details in Appendix B.1.

## 3.2 Game-theory optimal actions

We build a normative POMDP model for social behaviors using the chronic social conflict paradigm. In this section, we start with an approximation where each agent has all information about itself and its opponents. In that case, we use game theory to define the optimal actions for each agent.

We formalize our behavioral paradigm as a normal form game where, on each iteration, two agents have to decide simultaneously on what action $a$ to take. We defined the possible actions $a$ as "attack" or "defend". Depending on the actions $a_1$ and $a_2$ selected by the two agents respectively, they received rewards $r$ defined as follows. If both agents chose to "defend", no fight happened, leading to a zero reward $r_1 = r_2 = 0$ assigned to each agent. If both agents "attacked", the outcome of the game was defined by their strengths $s$, an additional parameter assigned to each agent in the model. The outcome probability $p^{win}$ was defined by the softmax rule over the strengths parameterized with the "outcome confidence" $\beta_o$:

$$p_i^{win} = Z^{-1} \exp(\beta_o s_i). \tag{1}$$

Here and below $Z$ denotes the normalization coefficient. Once the outcome was determined, the winning agent received a reward of $r = +1$, and the losing agent expended a cost of $r = -\mathcal{A}$. The reward expectation was equal to:

$$\kappa_i = 1 \cdot p_i^{win} + (-\mathcal{A}) \cdot (1 - p_i^{win}) = (1 + \mathcal{A})p_i^{win} - \mathcal{A}. \tag{2}$$

The cost of loss was reduced if one of the agents chose to "defend" while the other agent "attacked". In that case, the "attacking" agent always won and received the reward of $r = +1$ while the losing agent expended the cost of $r = -\alpha$ such that $\alpha < \mathcal{A}$. The reward expectations were described in the payoff matrix $\hat{R}_i$ whose rows correspond to the actions of the agent ("attack" and "defend") and columns correspond to the actions of its opponent:

$$\hat{R}_i = \begin{pmatrix} \kappa_i & 1 \\ -\alpha & 0 \end{pmatrix}. \tag{3}$$

To determine the optimal strategies in this game, we used game theory. In this approach, the goal of every participant was to maximize its expected reward $\mathbb{E}[r_i]$:

$$\mathbb{E}[r_1] = P_1^T \hat{R}_1 P_2. \tag{4}$$

Here the vectors $P_i$ define the probabilities to "attack" $p_i$ and to "defend" $1 - p_i$ for an agent $i$:

$$P_i \equiv \begin{pmatrix} p_i \\ 1 - p_i \end{pmatrix}. \tag{5}$$

A similar expression can be written for the expected reward of the second agent $\mathbb{E}[r_2]$. To maximize the rewards $\mathbb{E}[r_i]$, we computed their gradients with respect to the probabilities to "attack" $p_i$ (an agent could only update its own policy, but not that of the opponent). We used these gradients to update the policies leading to joint maximization of the expected rewards (Figure 2A):

$$\begin{cases} \frac{\partial \mathbb{E}[r_1]}{\partial p_1} = (\kappa_1 + \alpha - 1)p_2 + 1; \\ \frac{\partial \mathbb{E}[r_2]}{\partial p_2} = (\kappa_2 + \alpha - 1)p_1 + 1. \end{cases} \tag{6}$$

We provide the derivation for the equation above in Appendix C.1.1. The probabilities of actions can be only defined in the range $0 \leq \{p_1, p_2\} \leq 1$. Their evolution $\dot{p}_i \propto \partial \mathbb{E}[r_i]/\partial p_i$, governed by the equations above (red arrows in Figure 2A), converges to zeros or ones (blue arrows in Figure 2A) forming pure strategies (we show the lack of mixed strategies in Appendix C.1.2). To determine optimal pure strategies, we chose the strategies whose reward gradients (red arrows in Figure 2B) pointed outwards the $[0-1]$ interval for both agents. We represented the optimal policies via the tensor $A$ of probabilities for each possible action $a$ depending on the agent's strength $s_1$ and their opponent's strength $s_2$, averaging over all optimal pure strategies (Figure 2C):

$$A_{aij} = Pr(a_1 = a | s_1 = i, s_2 = j). \tag{7}$$

The resulting optimal policies in the model depended on the relative strengths of the agents (Figure 2C). We parameterized the optimal policies with the maximum relative strength of the agents $\delta = s_2 - s_1$ at which it was still optimal to "attack". We then explored how $\delta$ depends on the task parameters $\alpha$ and $\beta_o$ and found that most of these parameters' values correspond to the optimal strategy with $\delta = 0$ ("x" in Figure 2D), i.e. to "attacking" any opponent who is weaker or equal. We describe additional details regarding the acquisition of optimal policies in Appendix B.2 and summarize the procedure in Supplementary Algorithm 1.

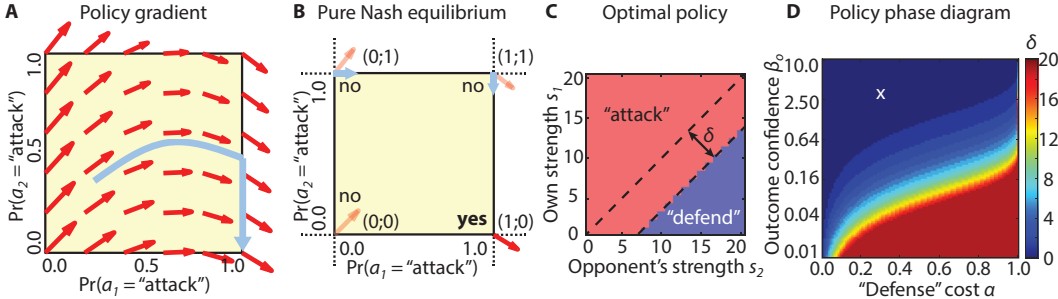

Figure 2: Game-theoretic model of the chronic social conflict paradigm. (A) Policy gradients. (B) Gradient orientations yield a pure strategy as a Nash equilibrium. (C) Optimal policy and its dependence on the relative strength. (D) Dependence of optimal policy on the model parameters.

## 3.3 Partial observability: beliefs about hidden variables

Game theory in our setting predicts a static policy, unchanging over time. Conversely, weight-matched mice in the experiment who initially attacked each other later split into always-attacking "winners" and ever-defending "losers" (Figure 1D). To account for this dynamic, we expand our model to a scenario where agents do not possess the full information about their strengths but accumulate this information over the task. In this section, we model the agents' information about strengths via beliefs – the probability distributions for an agent to belong to a certain strength category.

We used body weights $w$ of the animals as a proxy of their strengths $s$ (Andersson and Iwasa, 1996; Cooper et al., 2020):

$$w_i \propto s_i. \tag{8}$$

We modeled the animals' initial estimates of their body weights $\tilde{w}$ as normally distributed around true values $w$. Under this assumption, we reconstructed the probability distributions for animals' strengths (their initial beliefs) $B^i \equiv Pr(s = i|\tilde{w})$ using their body weights $\tilde{w}$ in Bayes' rule:

$$B^i \equiv Pr(s = i|\tilde{w}) = \frac{P_{\mathcal{N}(i,\sigma)}(\tilde{w})Pr(s = i)}{Pr(\tilde{w})}, \tag{9}$$

where $P_{\mathcal{N}(i,\sigma)}(\tilde{w})$ is a normal distribution probability density function representing the noise in the estimate of the animal's body weight; $Pr(\tilde{w})$ is the distribution of the animals' estimated body weights $\{\tilde{w}_i\}$, and $Pr(s)$ is the distribution of their strengths $\{s_i\}$. To denoise the strength distribution $Pr(s)$, we approximated the experimentally observed body weight distribution $Pr(w) \propto Pr(s)$ with a normal distribution.

We consider up to four types of beliefs. The two "primary" beliefs describe the animals' estimates of their own strength and of the strength of the opponent. The two "secondary" beliefs estimate the "primary" beliefs of the opponent. The animals were expected to better estimate their own strength compared to that of their opponents. To this end, we used separate uncertainty parameters for estimating one's own strength ($\sigma = \sigma_1$) and that of an opponent ($\sigma = \sigma_2$). For the "secondary" beliefs, the uncertainties were combined ($\sigma = \sqrt{\sigma_1^2 + \sigma_2^2}$). We provide the details regarding the initialization of beliefs in Appendix B.3 and summarize them in Supplementary Algorithm 2.

## 3.4 Evidence accumulation: Bayesian update of beliefs

In this section, we expand our model with an update mechanism for the agents' beliefs. We update the beliefs $B$ using Bayes' rule and the information about the actions $a$ and outcomes $o$ of agonistic interactions (Figure 3A).

We define the *outcome tensor* $O_{ij}^{oab}$ as a tensor describing the probabilities of each possible outcome $o_1 = o$ for an agent of a strength $s_1 = i$ after choosing an action $a_1 = a$ provided that the opponent has a strength $s_2 = j$ and chose an action $a_2 = b$:

$$O_{ij}^{oab} = Pr(o_1 = o|a_1 = a, a_2 = b, s_1 = i, s_2 = j). \tag{10}$$

The *reward tensor* $R_{ij}^{ab}$ describes the expected reward $\mathbb{E}[r_1]$ for an agent of a strength $s_1 = i$ after choosing an action $a_1 = a$ provided that the opponent has a strength $s_2 = j$ and chose an action

$a_2 = b$. The reward expectation was based on probabilities of possible outcomes:

$$R_{ij}^{ab} = \sum_o r(o, a, b)O_{ij}^{oab}.$$ (11)

The *outcome-action* and *reward-action tensors* predicted the probability of an outcome $o_1$ and expectation of a reward $r_1$ assuming that the opponent's actions $a_2$ are game-theory-optimal:

$$[RA]_{ijkl}^{a} \equiv \sum_b R_{ij}^{ab}\text{conv}_{kl}(A, P_{\mathcal{N}(0, \sqrt{\sigma_1^2 + \sigma_2^2})})_{kl}^b;$$ (12)

$$[OA]_{ijkl}^{oab} \equiv O_{ij}^{oab}\text{conv}_{kl}(A, P_{\mathcal{N}(0, \sqrt{\sigma_1^2 + \sigma_2^2})})_{kl}^b.$$ (13)

Here the opponent's action is estimated based on the "secondary" beliefs and the estimated outcome is based on the "primary" beliefs. The convolution reflects that the opponent's action is based on the distributional belief rather than on a point estimate of the strengths. The standard deviation $\sqrt{\sigma_1^2 + \sigma_2^2}$ applies to the estimated opponent's estimates of both its own strength and "our" strength.

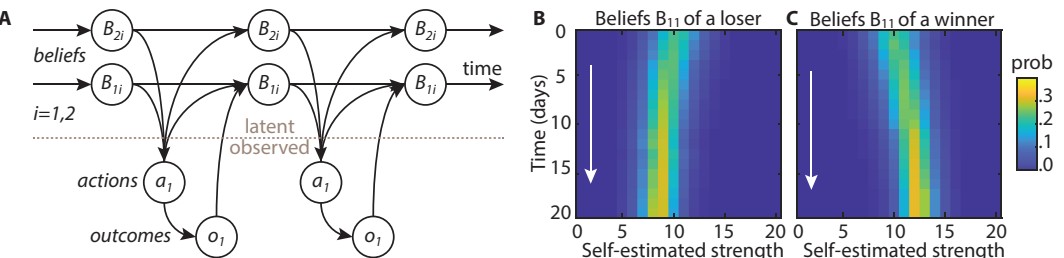

Figure 3: Bayesian update of beliefs in the model. (A) Belief update diagram for two mice. (B,C) Reconstructed belief dynamics for representative loser and winner mice.

To decide on an action $a_1$, an agent maximized its reward $r_1$. We computed this reward by summing the reward-action tensor $[RA]_{ijkl}^{a}$ multiplied by the belief tensors $B_{11}^i$, $B_{12}^j$, $B_{21}^k$, $B_{22}^l$ reflecting the probability distributions for strengths:

$$a_1 = \text{softmax}_a \sum_{ijkl} [RA]_{ijkl}^{a} B_{11}^i B_{12}^j B_{21}^k B_{22}^l.$$ (14)

Here the belief tensors $B_{11}$ and $B_{12}$ describe the "primary" beliefs of an agent about its own strength and the strength of its opponent respectively. Likewise, the belief tensors $B_{21}$ and $B_{22}$ describe the "secondary" beliefs, i.e. the agent's estimate of its opponent's "primary" beliefs. The indices $i, j, k, l$ iterate over all possible strengths, e.g. $B_{11}^5 = 0.2$ would mean that, according to the agent's belief, the probability of its own strength to be equal to 5 constitutes 20%. To update the agents' beliefs using the observed actions and outcomes $\{o_1, a_2\}$, we used Bayes' rule individually for every element of each belief tensor $B_{**}^i$:

$$Pr(s_1 = i|\{o_1, a_2\}) = \frac{Pr(\{o_1, a_2\}|s_1 = i)Pr(s_1 = i)}{Pr(\{o_1, a_2\})}.$$ (15)

The probability of an outcome $o_1$ and an action $a_2$ for the agent of the strength $s_1 = i$ was derived from the outcome-action tensor $[OA]_{ijkl}^{o_1 a_1 a_2}$:

$$Pr(\{o_1, a_2\}|s_1 = i) = \sum_{jkl} [OA]_{ijkl}^{o_1 a_1 a_2}.$$ (16)

The probability for the agent to be of strength $s_1 = i$ was taken from the belief tensor $B_{11}^i$:

$$Pr(s_1 = i) = B_{11}^i.$$ (17)

The marginal probability of the observation $\{o_1, a_2\}$ was defined by the outcome-action tensor $[OA]_{ijkl}^{o_1 a_1 a_2}$ scaled by the beliefs about strength $B_{11}^i$, $B_{12}^j$, etc. to scale the optimal actions with the probabilities of their underlying strengths:

$$Pr(\{o_1, a_2\}) = Pr(o_1|a_2)Pr(a_2) = \sum_{ijkl} [OA]_{ijkl}^{o_1 a_1 a_2} B_{11}^i B_{12}^j B_{21}^k B_{22}^l.$$ (18)

Together, the four equations above formed the update rule for agent beliefs based on their observations:

$$Pr(s_1 = i | \{o_1, a_2\}) = \frac{\sum_{jkl}[OA]_{ijkl}^{o_1 a_1 a_2} B_{11}^i B_{12}^j B_{21}^k B_{22}^l}{\sum_{ijkl}[OA]_{ijkl}^{o_1 a_1 a_2} B_{11}^i B_{12}^j B_{21}^k B_{22}^l}. \tag{19}$$

The update could proceed at an arbitrary learning rate $\varepsilon$:

$$B_{11}^i \leftarrow \varepsilon Pr(s_1 = i | \{o_1, a_2\}) + (1 - \varepsilon) B_{11}^i \tag{20}$$

We describe additional details regarding the update procedure for the beliefs in Appendix B.4 and summarize them in Supplementary Algorithm 3.

## 4 Results: inference of animal choice mechanisms in social conflict

### 4.1 Model fit and comparison

We used the framework defined above to test hypotheses about mouse social conflict-related choices. To test hypotheses against their alternatives, we optimized pairs of models on the training data (88 mice participating for 22 days). To this end, we specified a negative log-likelihood (NLL) function $\mathcal{L}$ comparing the action probabilities predicted in our model to the mouse actions logged in the experiment:

$$\mathcal{L} = -\sum_{mt}(\log Pr(a_m^t) + \log Pr(o_m^t)). \tag{21}$$

To fit the model parameters, we minimized the NLL regularized with the $l_2$ norm of the model arguments. We chose the regularization coefficient such that it resulted in the best fits in a simulated experiment. For the real mice in the experiment, we used the data spanning all 22 days to propagate the beliefs but the NLL was only computed for the data from days 1-3 and 21-22 to avoid the impact of the repeated actions on days 4-20. We report the implementation details of the model fit in Appendix B.5 and summarize them in Supplementary Algorithm 4.

We then proceeded to compare how well different models explain the behaviors observed in our experiment. Here we follow the standard way to compare models in behavioral research that is using formal comparison criteria, such as the Akaike Information Criterion (AIC) or the Bayesian Information Criterion (BIC). Both criteria are computed as a function of the number of parameters in the model less the log-likelihood of this model to generate the original data. Most of the models that we considered have the same number of parameters, therefore, as we only evaluated the difference of AICs (or BICs) between the models, the term reflecting the number of parameters cancelled out, resulting in the difference in log-likelihoods that we present.

To compare the models, we computed the changes in the NLLs based on the predictions of the models for the testing data (114 mice participating for 3/10/20 days). We performed the t-test on the changes in NLLs for individual mice followed by the false discovery rate (FDR) correction. Our comparisons were iterative: every time a model under consideration outperformed the previous best model, it became the new best model, starting a new round of comparisons. This procedure lasted until a model under consideration could not be outperformed by any other model. We used the q-value of $q = 0.05$ as a cutoff. Below, we report the results for the final round of the comparisons. We report additional details in Appendix B.6; the results of simulated experiments are described in Appendix C.2.

#### 4.1.1 Baseline models

Here we use the conventional approach and compare our model with the standard baselines (Devaine et al., 2014; Khalvati et al., 2019). The comparison results are displayed in Figure 4. We provide detailed numerical results in Supplementary Table 7 in Appendix C.3.

First, we determined the **depth of reasoning** involved in social conflict. We found that the Bayesian model that, along with "primary" beliefs, uses "secondary" beliefs to anticipate opponents' actions (a first-order theory of mind, 1-ToM) is more consistent with the data than a model where decisions are only based on "primary" beliefs about the strengths of the contestants (0-ToM).

Second, we analyzed the **availability of information** during conflict behavior. We established that the model that endows each animal with individual beliefs explains the animals' actions in the experiment better than the model that uses "shared" beliefs available to both animals in the pair.

Third, we assessed the **flexibility of policy** during a single agonistic interaction. We show that the model where animals decide on their actions before a fight ("fixed policy") explains our data better than the model where animals adapt their actions to the opponent's actions during a fight thus converging to a different Nash equilibrium ("flexible policy"; Appendix C.1.3).

Finally, to probe the **class of algorithms** used by animals in social conflict, we compared the Bayesian belief-based model with the Rescorla-Wagner reinforcement learning model where each action ("attack", "defend") was associated with a value updated based on the rewards. Our behavioral data was insufficient to distinguish between the two algorithms; we present additional considerations based on neural data in Section 4.2.

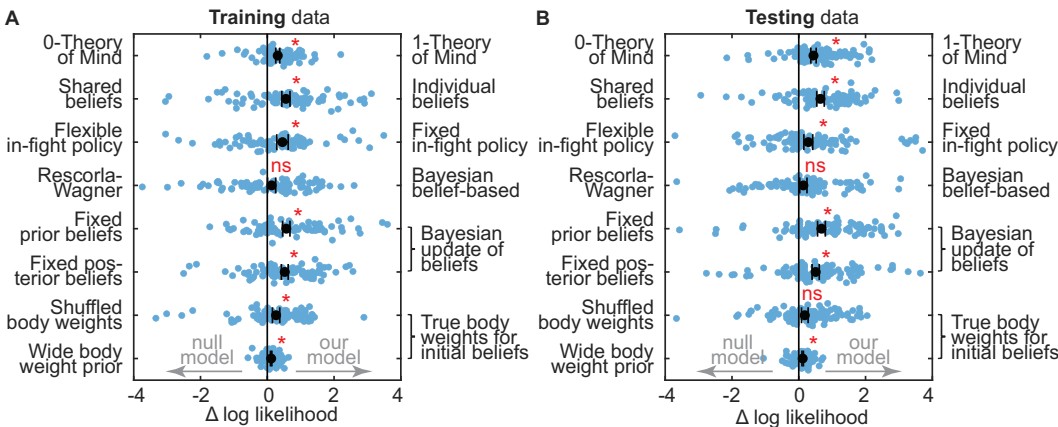

Figure 4: Model comparison on (A) training data and (B) testing data. Here (*) indicates a significant difference between the models (FDR $q \leq 0.05$), (ns) indicates a non-significant difference (FDR $q > 0.05$), and the whiskers show the mean $\pm$ the standard error of the mean.

### 4.1.2 Ablations

To infer the **dynamics of beliefs** that best describe the animals' actions, we compared the Bayesian belief-based model ("dynamic beliefs") with two models based on fixed beliefs ("static beliefs"; equivalent to the zero learning rate) (Baker et al., 2011). We found that the "dynamic beliefs" explained the animal data better than two types of "static beliefs": "prior beliefs" identical to those used to initialize the "dynamic beliefs" model and "posterior beliefs" identical to the output of the "dynamic beliefs" model reflecting the most complete knowledge about the animals.

To test the **role of body weight** in the initial beliefs about animals' strengths, we compared the model where the beliefs were initialized using the animals' body weights with the model where the body weights were shuffled and with the model where the correct body weights were used but their prior distribution was wider than the true distribution. Our comparison shows that using the true animals' body weights and their distribution to initialize beliefs has positively affected the predictions.

### 4.1.3 Optimal parameters

To obtain further insights into the aggressive behavior of mice, we evaluated the optimal parameters for the first-order Bayesian belief-based model. We observed a local minimum of the NLL at the parameter values $\sigma_1 = 3 \pm 1\text{g}$, $\sigma_2 = 10 \pm 1\text{g}$, $\beta_o = 6 \pm 1$, $\beta_a = 3 \pm 1$, $\alpha = 3 \pm 1$, $\mathcal{A} = 10 \pm 1$, and $\varepsilon = 0.8 \pm 0.2$. The identified parameters $\beta_o$ and $\alpha$ correspond to the policy with $\delta = 0$ where mice attack any opponents of equal or lower strength. The uncertainty $\sigma_1$ in initial estimates of own strength has a relatively low value suggesting that mouse body weight is an informative proxy for its strength. The uncertainty $\sigma_2$ in initial estimates of the opponent's strength is relatively high suggesting that the opponent's body weight carries no significant information about its estimated strength and that such strength is rather estimated based on their actions. We summarize these findings in a normative theory of social conflict in Appendix C.4.

## 4.2 Model correlates in the brain

To analyze the neural correlates of the model variables, based on the behavioral data, we estimated the beliefs for each mouse (Figure 5A-D). We then computed their correlations, voxel by voxel, with the c-Fos activity in the entire brain (Figure 5E). We evaluated the beliefs at the end of the experiments because the c-Fos imaging only allows collecting one brain activity snapshot per animal. To study brain activity at different stages of social conflict, we used mice with varied participation in the experiment, i.e. "winners" ($W$) and "losers" ($L$) on days 3/10/20 (subscripted in the group names in Figure 5) and also inverts (e.g. "winners" who become "losers", $WL$) on the day 22. We describe the details of this comparison in Appendix B.7 and summarize them in Supplementary Algorithm 5.

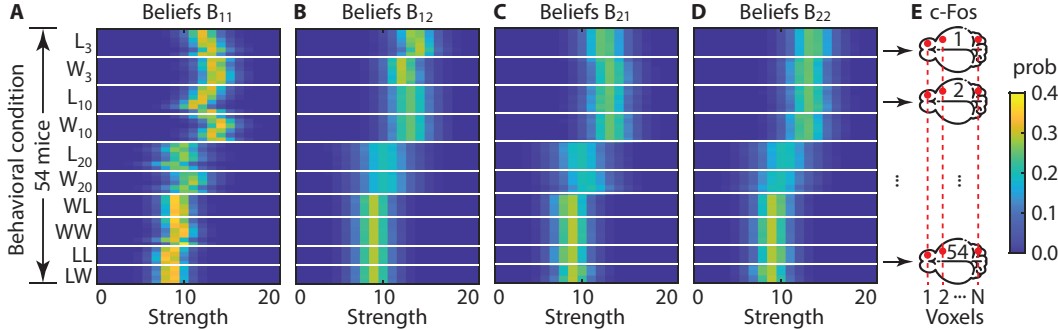

Figure 5: Reconstructed beliefs for individual mice: (A) about oneself; (B) about last opponent; (C) about last opponent's belief about oneself; (D) about last opponent's belief about themselves. Group key: $L_3$ stands for "losers" observed on day 3 etc. $WL$ stands for "winners" who become "losers" observed on day 22 etc. (E) Belief regression to the c-Fos activity in whole-brain samples.

In the correlation analysis, we use the "primary" and "secondary" beliefs reconstructed with the 0-ToM and 1-ToM models. We found significant (cumulative FDR $q \leq 0.1$) neural correlates for both types of these variables (Figure 6A-F). To analyze the representations of 0-ToM and 1-ToM beliefs, we examined the set of voxels whose activity was correlated with either 0-ToM or 1-ToM beliefs (Figure 6G), i.e. the union of the voxels correlated with the two models. We found that 77% of these voxels were correlated with the 1-ToM beliefs only, while 8% of the voxels were uniquely correlated with the 0-ToM beliefs. The remaining 15% of voxels were correlated with both 0-ToM and 1-ToM beliefs. Thus, brain activity appears to contain signatures of both 1-ToM and 0-ToM beliefs, suggesting that both models may be relevant to the animals' behavior. We describe the results of additional tests, further supporting this conclusion, at the end of this section.

We evaluated the brain regions hosting significant neural correlates of 1-ToM beliefs. We found that the "primary" beliefs about oneself and the opponents were correlated with clusters of neural activity in the median preoptic nucleus (MEPO), the periventricular hypothalamic nucleus (PV), and the parabrachial nucleus (PB). The "secondary" beliefs were correlated with neural activity in the brain regions including the medial septal nucleus (MS), the medial and lateral preoptic areas (MPO, LPO), the lateral septal nucleus (LS), the superior and inferior colliculi (SC, IC), the dentate gyrus (DG), the parabrachial nucleus (PB), the anterodorsal and median preoptic nuclei (ADP, MEPO), the periventricular hypothalamic nucleus and the dorsomedial nucleus of the hypothalamus (PV, DMH), the pallidum (PAL), the zona incerta (ZI), the tuberomammillary and tuberal nuclei (TM, TU), the pontine reticular nucleus (PRN), and the lateral hypothalamic area (LHA). Some of these regions are known for their involvement in conflict behaviors (Aleyasin et al., 2018; Diaz and Lin, 2020).

To control for **alternative explanations** of the observed neural activity, we repeated the analysis for model-unrelated variables. These variables included the body weights of the animals (used to initialize the beliefs), the outcomes of agonistic interactions, and the binary winner/loser variable. As a control, we also considered the shuffled beliefs of each type, such that we shuffled the animal identities within all groups of "winners" or "losers". Among these, we only found substantial correlates of the winner/loser variable whose presence in the brain is unsurprising. The correlates of this variable only partially overlapped with the correlates of the beliefs (Figure 6G). We further examined the neural correlates of variables reconstructed with the Rescorla-Wagner model. We found that such correlates are mostly explained by the winner/loser variable, whereas a larger portion of

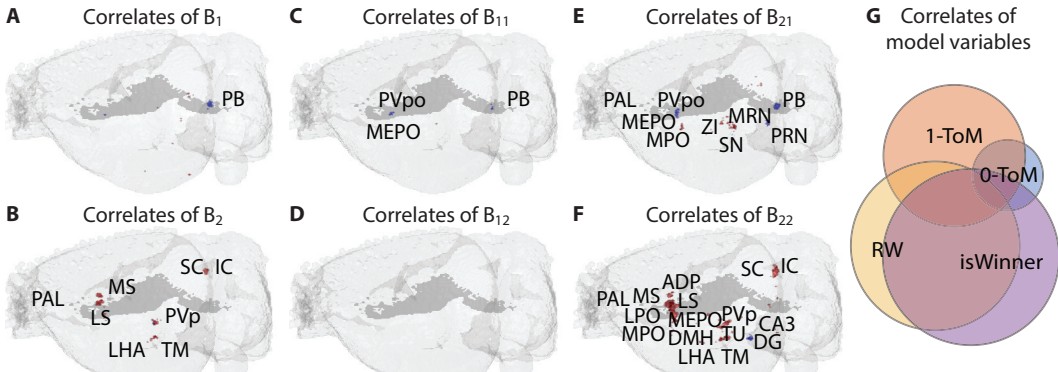

Figure 6: Correlates of the reconstructed beliefs in the c-Fos activity in the brain. (A-B) 0-ToM; (C-F) 1-ToM. Red: positive correlation; blue: negative correlation. (G) Voxels correlated with the models' latent variables.

the 1-ToM correlates does not have an alternative explanation (Figure 6G). This result supports the relevance of the proposed ToM model for conflict-related decision-making.

## 5 Broader impact

In our model, we applied evidence accumulation approaches to the domain of game theory which studies optimal interactions between agents. This allowed us to build a theory of chronic social conflict that can be used in future works for building quantitative models of social interactions. We then used Inverse Rational Control (IRC) to model the beliefs of animals based on their behavior. Combining the IRC with evidence accumulation models limits its degrees of freedom, increases robustness, and offers an interpretation for its predictions. Finally, we used the whole-brain c-Fos data (a proxy of neuronal activity) in combination with the IRC for an unbiased search for neural correlates of reconstructed beliefs. Overall, we combine game theory, evidence accumulation models, inverse rational control, and whole-brain imaging to propose a framework for building normative models of social behaviors and grounding them in neural circuitry in the brain. All animal procedures in this work were approved by the Stony Brook University Institutional Animal Care and Use Committee in accordance with the National Institutes of Health regulations. A detailed discussion of our results including the **strengths and limitations** of our approach is provided in Appendix A.

## Acknowledgements

We thank Pavel Osten and Kannan Umadevi Venkataraju for collecting and processing the imaging data; Nikhil Bhattasali and Khristina Samoilova for helpful discussions. This work was supported by grants from the National Institutes of Health (R01 DA050374; R01 AG076937), RSF (19-15-00026 to NK), the Swartz Foundation for Computational Neuroscience, and the Mathers Foundation.

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

# A Discussion

## A.1 Normative theory of social conflict

In this work, we formulated a theory of social conflict based on game theory and Bayesian inference of agents' beliefs. We refined our model using mouse behavioral data and further validated it using neuronal data.

Our results suggest that animals' behavior during conflict is consistent with maintaining and updating beliefs about their strength and the strengths of their opponents. This observation is similar to prior work on Bayesian Theories of Mind proposing that in a variety of tasks humans and animals make decisions based on estimates of environmental variables updated with Bayes' rule (Baker et al., 2011). This finding also supports prior work showing the lack of naive (without evolving beliefs) game-theory-optimal behaviors in humans (Stahl and Wilson, 1995).

Our data suggest that, along with "primary" beliefs about their own and opponent's strengths, animals maintain "secondary" beliefs estimating the "primary" beliefs of opponents. This result is consistent with previous work in humans reporting that choices in the volunteer's dilemma are better explained with the 1-ToM model compared to 0-ToM and 2-ToM models (Khalvati et al., 2019). Our results are also consistent with behavioral observations in mice suggesting that animals facing an aggressive/defeated opponent anticipate their actions (Kudryavtseva et al., 2014).

We show that initial estimates of animals' own strength were correlated with their body weight. This finding is consistent with the proposals that body weight is a strong predictor for fighting performance in mice (Andersson and Iwasa, 1996; Cooper et al., 2020) and social status in rats (Nagy et al., 2023). At the same time, our data suggest that animals may not use correlates of opponents' body weights as surrogates for strength, rather relying on opponents' actions.

Our fits to behavioral data suggest that the decisions about animals' actions in the chronic conflict task were made before each agonistic interaction with other mice and were not optimized within a single interaction. These fits are consistent with our observations of animal behavior in individual interactions. This result also aligns with prior work in humans reporting bounded rationality in a series of games (Stahl and Wilson, 1995). Further experiments may be designed to validate this result.

Using behavioral data, we have arrived at a payoff matrix describing the animals' decisions in social conflict. We found that the cost of defense (via escaping confrontation) is small compared to the cost of defeat (via attack), in agreement with minimizing the physical damage reported by Crowcroft (1966). At the same time, the cost of defeat after an attack exceeds the reward for victory. This may be due to the immediate effect of the cost of defeat (e.g. physical damage) compared to a discounted delayed reward (e.g. mating, access to food, etc.) associated with winning in a conflict (Kable and Glimcher, 2007; Kobayashi and Schultz, 2008).

Finally, we identified the neural correlates of the beliefs computed in our model. Previous studies have characterized multiple brain regions whose activity modulates aggression. Our approach offers an interpretation of the computations performed in several brain regions. We find correlates of beliefs in hypothalamic regions, consistent with the previously reported role of the hypothalamus in aggression (Aleyasin et al., 2018; Diaz and Lin, 2020). Within the hypothalamus, a recent study describes the *short-term* aggression ramping encoded in the ventromedial hypothalamus (VMHvl) but not in the medial preoptic area (MPO) (Nair et al., 2023). Here, we observe the correlates of animals' beliefs in preoptic areas (MPO, LPO, MEPO) but not in VMHvl in the conditions of *chronic* social conflict. This finding may indicate differences in the coding schemes for short-term and chronic conflict. We find that the neural activity in MPO is correlated with the "secondary" beliefs about opponents' strengths, in agreement with a recent study on the role of cMPO neurons (Wei et al., 2023). It is likely that other brain regions, such as VMHvl encode variables different from beliefs. This possibility may explain why we did not find significant correlates of beliefs in other brain regions related to aggression including some limbic areas, the brain stem, and the reward system (Aleyasin et al., 2018).

## A.2 Limitations and strengths of our approach

In our setting, animals made their decisions in the conditions of incomplete information and refined their policies based on the observed outcomes of their encounters. Agents in such a setting face the problem of a partially observable Markov decision process (POMDP). Here, we use biological

decision-making to propose an approach to solving the POMDP problem in multi-agent settings. Our work extends machine-learning methods for analyzing behavioral and neural data in these conditions. Combining inverse rational control (Dvijotham and Todorov, 2010; Kwon et al., 2020) with game theory (Cressman et al., 2003) offers an interpretation of model parameters. Specifically, the variables in our models either represent the rewards for various outcomes or describe the confidence in taking various actions. A low number of fitting parameters makes our approach data-efficient. This property may be useful in neuroscience where data are typically small (Stahl and Wilson, 1995). Game theory makes the transitions between beliefs deterministic, reducing the computations compared to probabilistic Baum-Welch algorithms (Baum et al., 1970), conventionally used in POMDPs (Wu et al., 2018). The benefits of our approach can be quantified in follow-up studies.

A limitation of our game-theory-based approach is that it involves model comparisons between *ad hoc* models (Stahl and Wilson, 1995). To mitigate this limitation, we based our models on the results of previous studies, using standard ToM/ML baselines (Stahl and Wilson, 1995; Khalvati et al., 2019). In future studies, we may consider additional models including mixtures of ToMs (Khalvati et al., 2019), naive Nash, and rational expectation strategies (Stahl and Wilson, 1995).

Our behavior-based model reconstructs relevant variables for individual animals, allowing us to search for their correlates in neural activity. As a proxy of neural activity, we used the c-Fos expression, captured with whole-brain 3D microscopy (Renier et al., 2016). Using c-Fos comes with several limitations: i) c-Fos lacks temporal resolution, allowing one snapshot of brain activity per mouse; ii) the c-Fos signal is nonspecific, potentially encoding factors other than neuronal activities (Herrera and Robertson, 1996). Among the benefits, c-Fos does not perturb animals' social behavior, making it the standard choice for studies of aggression (Aleyasin et al., 2018; Diaz and Lin, 2020; Wei et al., 2021). *Ex-vivo* 3D imaging allowed us to reconstruct the whole-brain neural activity. While previous studies focused on tissue sections of preselected brain regions, our whole-brain data has enabled an unbiased study of conflict circuitry.

The neural activation analysis in this work is performed with the correlation analysis followed by the FDR correction (Benjamini and Hochberg, 1995). This is a conservative analysis establishing whether the model variables are directly represented in neural activity. Alternative approaches could include GLMs for detecting linear combinations of model parameters (Lindquist, 2008), Gaussian random fields for modeling spatial correlations of the signal (Rue and Held, 2005), and MANOVA for distinguishing representations of correlated model variables (Allefeld and Haynes, 2014). Using these and other fMRI approaches (Skup, 2010; Ashburner et al., 2014) may allow identifying the correlates of individual model variables.

Our results are based on data comprising "attack"/"defend" and "win"/"lose" readouts but our analyses can be extended to different metrics and social behaviors. Long-term tracking of social structure in rats (Nagy et al., 2023) may allow for testing our conclusions in a different species in a complex social environment. The data on carrier/non-carrier phenotypes (Schroeder and Desor, 2005) would allow us to see if our results generalize to a cooperative setting. The model may also be applied to other domains, such as automated negotiation (Baarslag et al., 2016). Overall, our framework may be used for studying competitive or cooperative behaviors in real-world settings using limited data.

# B   Methods

## B.1   Mouse chronic social conflict paradigm

To induce varied behavioral states in mice, we applied the chronic social conflict paradigm. Pairs of weight-matched mice were separated by a perforated partition in cages. Once daily, the partition was removed for 10 minutes to enable agonistic interactions between mice. To study the adaptive properties of the aggressive/defeated states in mice, we kept the opponents unchanged for 3 days. Then, to reveal the pathological properties of behavioral states, each winning mouse was kept in its cage, while each losing mouse was daily relocated to an unfamiliar cage with an unfamiliar winner. Non-fighting mice were discontinued from participating in the experiment. After a total of 20 days of interactions, to reveal the flexibility of behavioral states, winning and losing mice were reorganized to face opponents of the same state. The newly formed pairs underwent 2 more days of interactions. During all interactions, we logged the mouse actions ("attack", "defend") and the outcomes of interactions ("win", "lose", "draw"). We varied the participation of mice in the experiment:

Table 1: Datasets

| DATASET | I | II | III | TOTAL |
|---|---|---|---|---|
| DAYS | 3 / 10 | 20 | 22 | |
| MOUSE BEHAVIOR | EXP: 56; CTL: 6 | EXP: 58; CTL: 6 | EXP: 88; CTL: 6 | EXP: 202; CTL: 18 |
| BRAIN 3D DATA | EXP: 24; CTL: 6 | EXP: 11; CTL: 6 | EXP: 19; CTL: 6 | EXP: 54; CTL: 18 |

## B.2 Game theory optimal actions

To define a class of models to test against our data, we described the mouse task as a normal-form game. Here, the agents simultaneously chose their actions; the available actions were to "attack" and to "defend". The outcome of the game depended on the actions and strengths of participating agents. The strengths $s_i$ were defined as constants between $1 - 20$, specific to an agent and unchanging.

Table 2: Game outcomes

| ACTION | #2 "ATTACKS" | #2 "DEFENDS" |
|---|---|---|
| #1 "ATTACKS" | $p_{win}^1 \sim Gibbs(s_1, s_2)$ | #1 WINS |
| #1 "DEFENDS" | #2 WINS | DRAW |

If both agents in the game chose to "attack", the probability of "winning" the game was defined by the softmax rule over their strengths parameterized with the "outcome confidence" $\beta_o$. The reward assigned to each agent depended on the outcome of the game and on their action as follows:

Table 3: Rewards

| ACTION | "WIN" | "LOSE" | "DRAW" |
|---|---|---|---|
| "ATTACK" | 1 | $-\mathcal{A}$ | N/A |
| "DEFEND" | N/A | $-\alpha$ | 0 |

To derive the optimal actions for agents in a fully observable setting, we parametrized the agents' actions by the probability to "attack" and derived the gradients of the expected reward w.r.t. these probabilities. We found pure-strategy Nash equilibria by choosing the pure strategies whose gradients were directed outside the support of action probabilities. We averaged the action probabilities over Nash equilibria. Finding the optimal policies is summarized in Algorithm 1:

---

**Algorithm 1** Game theory optimal actions

---

**Input:** strength range $s_{max}$, costs $\alpha$, $\mathcal{A}$, confidence $\beta_o$
Initialize actions $A = \text{zeros}(2, s_{max}, s_{max})$.

**for** strengths $s_1, s_2 = 1$ **to** $s_{max}$ **do**
    expectation $\kappa_1 = (1 + \mathcal{A}) \text{softmax}_s(\beta_o s_1, \beta_o s_2) - \mathcal{A}$
    gradient $\dot{p}_1 = (\kappa_1 + \alpha - 1)p_2 + 1$; similarly find $\dot{p}_2$
    **for** policies $p_1, p_2 = 0$ **or** 1 **do**
        Find pure strategy: $\dot{p}_{1(2)}(p_{1(2)} - 0.5) > 0$
    **end for**
    Average the strategies: $A(1, s_1, s_2) = \text{mean}(\{p_1\})$
    $A(2, s_1, s_2) = 1 - A(1, s_1, s_2)$
**end for**

---

## B.3 Bayesian belief initialization

To account for the partial observability of information in the task, we defined the probability distributions (beliefs) describing the strengths of the agents. We initialized the beliefs with the normal distributions whose mean values corresponded to the body weight of the animals (shifted by -15g to fit the range). To account for the prior information about the body weight distribution, we applied

Bayes' rule to the initial beliefs. To reflect possible depths of reasoning, we considered four types of beliefs parameterized by different standard deviations of the normal distribution:

Table 4: Initial belief standard deviations

| Belief | "Mine" | "Opponent's" |
|---|---|---|
| "about myself" | $\sigma_1$ | $\sqrt{\sigma_1^2 + \sigma_2^2}$ |
| "about opponent" | $\sigma_2$ | $\sqrt{\sigma_2^2 + \sigma_1^2}$ |

The "opponent's" beliefs here reflect the agent's beliefs about the opponent's beliefs that may differ from the opponent's beliefs in the model. Overall, the acquisition of initial beliefs about the animals' strengths is summarized in Algorithm 2. Here $B^i_{m_1 m_2}$ is the belief of mouse $m_1$ about mouse $m_2$.

---

**Algorithm 2** Bayesian belief initialization

---

**Input:** body weights $w_i$, uncertainties $\sigma_1, \sigma_2$
Initialize beliefs $B = \text{zeros}(s_{max}, \max(m), \max(m))$.
strengths $Pr(s) = P_{\mathcal{N}(\mathbb{E}[w], \mathbb{D}[w])}$

**for** mice $m_1, m_2 = 1$ **to** $\max(\{m\})$ **do**
  Set $\sigma$ using Table 4
  **for** strength $i = 1$ **to** $\max(\{s\})$ **do**
    belief $B^i_{m_1 m_2} = Z^{-1} P_{\mathcal{N}(i,\sigma)}(w_{m_2}) Pr(s)(i)$
  **end for**
**end for**

---

### B.4 Bayesian belief update

To enable evidence accumulation in our model, we used Bayes' rule to update the beliefs based on the observations. The agonistic interaction outcomes were used to update the primary beliefs about the agent's own and the opponent's strengths. The opponents' actions were used to update the beliefs about the opponent's beliefs. The update procedure for beliefs is summarized in Algorithm 3 below.

---

**Algorithm 3** Bayesian belief update

---

**Input:** pairs $p^t_m$, actions $a^t_m$, outcomes $o^t_m$
Compute actions $A$ using Algorithm 1; precompute
$[RA]^a_{ijkl} \equiv \sum_b R^{ab}_{ij} \text{conv}_{kl}(A, P_{\mathcal{N}(0,\sqrt{\sigma_1^2+\sigma_2^2})})^b_{kl}$
$[OA]^{oab}_{ijkl} \equiv O^{oab}_{ij} \text{conv}_{kl}(A, P_{\mathcal{N}(0,\sqrt{\sigma_1^2+\sigma_2^2})})^b_{kl}$
Initialize beliefs $B$ using Algorithm 2

**for** time $t = 1$ **to** $\max(\{t\})$ **do**
  **for** mouse $m_1 = 1$ **to** $\max(\{m\})$ **do**
    opponent $m_2 = p^t_{m_1}$
    **for** mouse **in** $m_1, m_2$ **do**
      $\tilde{a}^t_k = \text{softmax}_a \sum_{ijkl} [RA]^a_{ijkl} B^i_{11} B^j_{12} B^k_{21} B^l_{22}$
    **end for**
    **for** strength $i = 1$ **to** $\max(\{s\})$ **do**
      **for** mice **in** $m_1, m_2$ **do**
        $\Delta B^i_{12} = Z^{-1} \sum_{jkl} [OA]^{o_1 a_1 a_2}_{ijkl} B^i_{11} B^j_{12} B^k_{21} B^l_{22}$
        update $B^i_{12} \leftarrow \varepsilon \Delta B^i_{12} + (1 - \varepsilon) B^i_{12}$
        Repeat for other belief types $B_{**}$
      **end for**
    **end for**
  **end for**
**end for**

---

## B.5 Model fit

Besides the belief update, the above algorithm allows for predicting the animals' actions based on their beliefs. We consider the opponent's predicted action to be game-theory-optimal based on the opponent's estimated beliefs. To predict the agent's action, we maximize its reward expectations based on its own "primary" beliefs and the opponent's predicted action. To optimize the model predictions, we update the model's parameters to minimize the negative log-likelihood (NLL) for the actions in our behavior data to be sampled from the model predictions. While we use the entire data to propagate the beliefs, we only use days 1-3 and 21-22 for the NLL evaluation to avoid the impact of repeated actions on days 4-20. We use bootstrap to compute the error quantiles for the identified parameters. Overall, the inference of model parameters is summarized in Algorithm 4 below.

---

**Algorithm 4** Model parameter inference

---

  **Input:** pairs $p_m^t$, actions $a_m^t$, outcomes $o_m^t$
  Initialize parameters $x \equiv [\sigma_1, \sigma_2, \beta_a, \beta_o, \alpha, \mathcal{A}, \varepsilon]$
  Initialize $x_{opt} = zeros(100, 7)$

  **for** repeat $i = 1$ **to** 100 **do**
      Define subset $\tau \in [1, T]$ with repetitions
      Define actions $Pr(a_m^\tau)$ using Algorithm 3
      Define likelihood $\mathcal{L} = -\sum_{mt} \log Pr(a_m^\tau)$
      Regularize $\mathcal{L} \leftarrow \mathcal{L} + \lambda \|x/x_{\max}\|_2$
      Optimize $x_{opt}(i) = fminsearch(\mathcal{L})$
  **end for**
  Compute $\mathbb{E}[x_{opt}], \mathbb{D}[x_{opt}]$

---

To optimize the parameter inference, we applied it to simulated data with known parameters. We added noise to Algorithm 2 to generate the initial beliefs about strengths in the model and used Algorithm 3 to simulate the actions. Once the model parameters were reconstructed with Algorithm 4, we used Gaussian processes to estimate the means and the error quantiles for the predictions. We varied the regularizer $\lambda \in \{0; 0.1; 1; 10\}$ and chose the one that led to the best reconstruction ($\lambda = 1$).

## B.6 Model comparison

To test hypotheses about mouse decision-making, we performed the model comparison. To this end, we optimized pairs of models on the training data (88 mice participating for 22 days) and computed the changes in the NLL based on the predictions for the testing data (104 mice participating for 3/10/20 days). We performed the t-tests on the changes in the NLLs for individual mice followed by the FDR correction. For pairwise comparisons, we picked an initial model and considered one-step deviations from it (e.g. for the 0-ToM model, we separately considered "frozen" prior beliefs and, alternatively, shuffled body weights, but not their combination). We fitted each model in the same way, using a numerical optimizer until convergence. We considered new models until one of the models outperformed the initial one; we then used this new model as an initial model for the next round of comparisons using one-step deviations from this new model. We stopped after arriving at a model that was not outperformed by any of the other models. Here we list the considered models and specify their differences in comparison with the "default" (1-ToM) model described above:

Table 5: Baseline models

| TEST | BASELINE MODEL |
|---|---|
| DEPTH OF REASONING | 0-THEORY OF MIND |
| AVAILABILITY OF INFORMATION | SHARED BELIEFS |
| FLEXIBILITY OF POLICY | FLEXIBLE IN-FIGHT POLICY |
| CLASS OF ALGORITHMS | RESCORLA-WAGNER |

In **Rescorla-Wagner** model, each action ("attack", "defend") was associated with a value representing the expected future reward. This value was independent of the mouse's strength or the identity of an opponent. The values for the "attack" and "defend" actions were initialized using the expectations

over the distribution of game theory optimal opponents, then updated at a learning rate $\varepsilon$ using the reward as a teaching signal. In the **0-Theory of Mind** model, we only considered the "primary" beliefs of an agent (about its own and the opponent's strength) assuming that the opponent acts in accordance with the same beliefs (even though the actual beliefs of the two agents were independent and may have differed). In the **shared beliefs model**, the 0-ToM beliefs were identical between the two opponents. In the **flexible in-fight policy** model, the policy gradients used in determining the Nash equilibria were computed for the expected rewards integrated with the "primary" beliefs of each mouse reflecting the in-fight adaptation to the opponent's actions. As the opponent's actions were observable in this setting, the model did not include "secondary" beliefs.

Table 6: Ablation studies

| TEST | ABLATION |
|---|---|
| DYNAMICS OF BELIEFS | FIXED PRIOR BELIEFS |
| | FIXED POSTERIOR BELIEFS |
| ROLE OF BODY WEIGHT | SHUFFLED BODY WEIGHTS |
| | WIDE BODY WEIGHT PRIOR |

In the **fixed prior beliefs** model we used the zero learning rate for the belief update, retaining the beliefs unchanged from their body weight-based initial values. In the **fixed posterior beliefs** model, we used the zero learning rate but the initial beliefs were substituted with the final beliefs produced with our default model. In the **shuffled body weights** model, we shuffled the animals' body weights before using them to initialize the beliefs. In the **wide body weight prior** model we used the double standard deviation for the prior distribution of the animals' body weights at the belief initialization.

## B.7 Neural correlates of model variables

To find neural correlates of the model variables, we normalized the registered c-Fos activity in each brain sample by dividing it by the average c-Fos activity of all control samples from the same dataset. We converted the relative c-Fos activities to the log scale to obtain a roughly normal distribution of relative activities. We normalized the model variables by their average estimated values for the control samples. Then, we computed the voxel-wise correlations between the processed c-Fos activity and model variables. To select the voxels significantly correlated with model variables, we performed the FDR correction. Finding the belief correlates in the brain is described in Algorithm 5 below.

---

**Algorithm 5** Belief correlates in the brain

---

**Input:** 3D brain images $I$ aligned to atlas $A$; beliefs $B$

**for** sample $s$ **in** experimental samples **do**
    Normalize $I_s = \log I_s - \log(\text{mean}(I_{ctrl}))$
    Normalize $b_s = \text{mean}(b_s) - \text{mean}(b_{ctrl})$
**end for**

**for** belief $b$ **in** $\{B_{11}, B_{12}, \text{etc.}\}$ **do**
    **for** voxel $i$ **in** 3D brain atlas volume **do**
        Compute $p_i \leftarrow \text{corr}(\{I\}_i, b)$
    **end for**
    Compute $\{q_i\} = \text{FDR}(\cup_i\{p_i\})$
    Find $corr_b = \{j \le i\} : \{\sum_{k=0}^{i} \text{sort}(q_k) < 0.1\}$
    Find brain regions $r$ in atlas $A$: $corr_b \in r$
**end for**

---

Considered variables included: "primary" and "secondary" beliefs about the mouse's own and the opponent's strength for 0-ToM and 1-ToM models; the estimated advantages (i.e. the differences between the same-order beliefs within a model); the estimated outcomes (i.e. the binarized estimated advantages); the predicted actions for oneself and the opponent; the Rescorla-Wagner values of the two actions and their difference. The control variables included: the body weights of mice, the observed actions, the outcomes of the agonistic interactions, and the shuffled beliefs of each type.

## B.8 Software and time budget

All computations were performed with Matlab R2022b. The mouse data fits were performed with a 3 GHz Intel Core i7-based Dell laptop computer. The simulated data fits were performed with a 3 GHz Intel Xeon-based Supermicro computer server in 40 parallel threads. 0-ToM models converged within 2 minutes (per model per thread); 1-ToM models converged within 3 hours (per model per thread).

## C Supplementary results

### C.1 Game theory optimal actions

#### C.1.1 Pure strategies

In this section, we provide a simple derivation for the game theory optimal actions in our model. The expected reward $\mathbb{E}[r_1]$ for the first agent can be computed as its payoff matrix $\hat{R}_1$ multiplied by the action probability vectors $P_{1,2}$ describing the policies of the first and the second agents:

$$\mathbb{E}[r_1] = P_1^T \hat{R}_1 P_2 = \begin{pmatrix} p_1 & 1-p_1 \end{pmatrix} \begin{pmatrix} \kappa_1 & 1 \\ -\alpha & 0 \end{pmatrix} \begin{pmatrix} p_2 \\ 1-p_2 \end{pmatrix} = p_1 p_2 (\kappa_1 + \alpha - 1) + p_1 - \alpha p_2. \quad (22)$$

Here $p_{1,2}$ are the probabilities to "attack" for the first and the second agents, $\kappa_1$ is the expected reward for the first agent in the case when both agents "attack", and $-\alpha$ is the cost for the first agent in the case when only the second agent "attacks". The first agent may increase its expected reward by changing its policy $P_1$ that depends on $p_1$ (the first agent can't change $p_2$ as $p_2$ is controlled by the opponent). To optimize the expected reward $\mathbb{E}[r_1]$, we compute its gradient w.r.t. $p_1$:

$$\frac{\partial \mathbb{E}[r_1]}{\partial p_1} = (\kappa_1 + \alpha - 1)p_2 + 1. \quad (23)$$

The agents may follow a pure strategy with binary probabilities $p_{1,2}$ to "attack". Such strategies correspond to Nash equilibria if the reward gradients point outwards the $[0-1]$ interval for both agents so that the policy won't have room to evolve. This requirement can be formalized as:

$$\begin{cases} \frac{\partial \mathbb{E}[r_1]}{\partial p_1}(p_1 - \frac{1}{2}) > 0; \\ \frac{\partial \mathbb{E}[r_2]}{\partial p_2}(p_2 - \frac{1}{2}) > 0. \end{cases} \quad (24)$$

Using the values of the derivatives from Equation 23, we arrive at the criterion for Nash equilibria:

$$\begin{cases} (p_2^{pure}(\kappa_1 + \alpha - 1) + 1)(p_1^{pure} - \frac{1}{2}) > 0; \\ (p_1^{pure}(\kappa_2 + \alpha - 1) + 1)(p_2^{pure} - \frac{1}{2}) > 0. \end{cases} \quad (25)$$

An arbitrary task setting (defined by the parameters $\kappa$ and $\alpha$) may correspond to multiple Nash equilibria. To this end, we approximate the expected policy by averaging the action probabilities $p_{1,2}$ over the identified equilibria. A better way to account for multiple Nash equilibria is to integrate the action probabilities over their basins of attraction; this analysis can be added in future studies.

#### C.1.2 Lack of mixed strategies

To consider mixed strategies with nonbinary probabilities to take either of the available actions, assume that both agents in the game follow the gradients of their respective rewards to update their action probabilities. Following Equation 23 for these gradients, we arrive at the dynamical system:

$$\begin{cases} \frac{dp_1}{dt} = p_2(\kappa_1 + \alpha - 1) + 1; \\ \frac{dp_2}{dt} = p_1(\kappa_2 + \alpha - 1) + 1. \end{cases} \quad (26)$$

Any mixed strategy $\{p_1^{mixed}, p_2^{mixed}\}$ corresponds to an attractor of this dynamical system, i.e. a point $\{p_1, p_2\}$ where the time derivatives of both variables $p_1, p_2$ are equal to zero and the phase portrait of the system is shaped as a stable node or a stable spiral so that the agents' action probabilities could converge to this point. To analyze the dynamical system around the point of zero gradient, we linearize its dynamics via computing the Jacobian:

$$\frac{d}{dt}\begin{pmatrix} p_1^* \\ p_2^* \end{pmatrix} = \begin{pmatrix} 0 & \kappa_1 + \alpha - 1 \\ \kappa_2 + \alpha - 1 & 0 \end{pmatrix} \begin{pmatrix} p_1^* \\ p_2^* \end{pmatrix}. \quad (27)$$

To describe the phase portrait of this system, we compute the eigenvalues of the Jacobian:

$$\lambda_{1,2} = \pm\sqrt{(\kappa_1 + \alpha - 1)(\kappa_2 + \alpha - 1)}. \tag{28}$$

In the case of real eigenvalues $\lambda_{1,2}$, their product is negative implying that the system's dynamics form a saddle point. In the case of complex eigenvalues, the zero trace of the Jacobian implies that the system's dynamics form a limit cycle. In both cases, the system's dynamics do not converge to the point of zero gradient, rendering mixed strategies infeasible.

### C.1.3 Flexible in-fight policy

In the case where agents do not have perfect information about each other's strengths, their estimates of the Nash equilibria may be based on their beliefs about the strengths of the agents in the game. As these beliefs, generally speaking, are different between the contestants, their estimates of the Nash equilibria may also be mismatched. As such, these estimates cannot be considered as true Nash equilibria, requiring further evolution of the contestants' policies.

One way to account for this discrepancy is to update the policies during the fight. Although our tests suggest that such a mechanism is inconsistent with animals' behavior in the experiment and, as such, **is not used in our final model**, we provide its description for completeness.

Updating the policies during the fight is supposed to lead to a joint Nash equilibrium. This involves two simple additions to our previously described model: i) the reward expectations in the payoff matrices $\hat{R}_{1,2}$ are integrated over the "primary" beliefs about the participants' strengths, and ii) the optimization is performed jointly for both participating agents as their policies are observable to each other under the assumptions about this case.

Amending the payoff matrix $\hat{R}_1$ can be reduced to redefining its entry $\kappa_2$, the only one depending on the agents' strengths. An in-fight Nash equilibrium is only defined under the 0-ToM model: the "primary" beliefs are used to predict the odds of winning while the "secondary" beliefs, normally used for predicting the opponent's actions, are unnecessary as actions are directly observed:

$$\kappa_2^2 = \sum_{ij} \kappa_2(s_2 = i, s_1 = j)B_{22}^i B_{21}^j. \tag{29}$$

That is, the equilibrium policy $p_1$ for the first agent depends on the second agent's estimate of its expected reward $\kappa_2^2$ for the case when both agents "attack". The expression for the optimal pure strategies is then rewritten based on Equation 25 as follows:

$$\begin{cases} (p_2^{pure*}(\kappa_1^1 + \alpha - 1) + 1)(p_1^{pure*} - \frac{1}{2}) > 0; \\ (p_1^{pure*}(\kappa_2^2 + \alpha - 1) + 1)(p_2^{pure*} - \frac{1}{2}) > 0. \end{cases} \tag{30}$$

The equations above describe the in-fight Nash equilibrium which is **not a part of our model**. This analysis, however, may be useful in future studies of different social behaviors.

### C.2 Model fit

In data fits, we used the entire data to propagate the beliefs while we only considered days 1-3 and 21-22 for the NLL evaluation. This is because, on days 4-20, "winners" were always matched with "losers" predetermining the results of the interactions and reducing the NLL sensitivity to the model parameters. As a result, the model using days 1-3 and 21-22 has a local minimum (Figure 7B) at the identified parameters, while the model using days 1-22 has a plateau instead (Figure 7A).

To assess the reliability of our data fits we performed additional fits on simulated data. To this end, we replicated the mouse experiment in a simulated environment and generated 500 choice patterns using various predefined parameters. We sampled parameters uniformly from the ranges previously used for parameter fits in mouse data, selecting only the policies that resulted in a 30%-70% fraction of either action. We then evaluated the accuracy of reconstructed beliefs in both 0-ToM (Figure 7C,D) and 1-ToM (Figure 7E-H) models.

### C.3 Model comparison

Below, we report the numerical results for the final round of model comparisons on our behavioral data. The differences in NLLs were computed for each mouse individually; here we show the mean

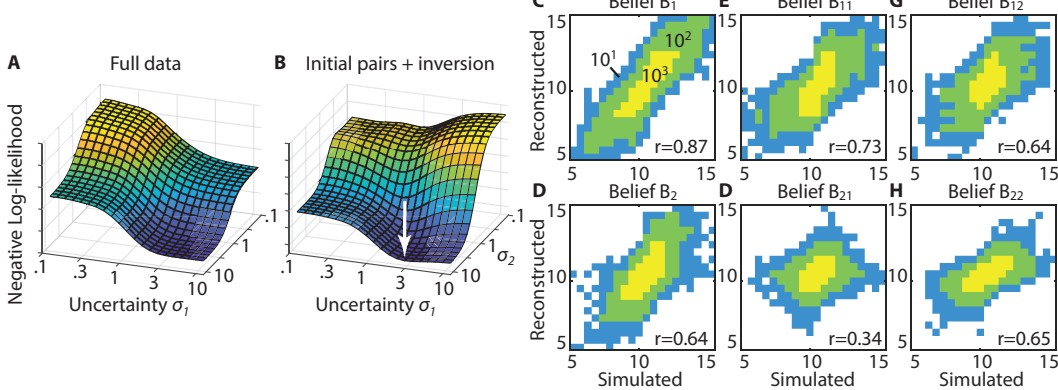

Figure 7: Parameter inference. (A-B) NLL for reconstructing the animal data; (C-H) reconstructed beliefs in simulation (color code: density of beliefs; r-values: Pearson correlation between simulated, reconstructed beliefs.

$\Delta NLL$ and its standard error of the mean for the training and testing datasets. Positive numbers correspond to our model explaining the data better than baselines/ablations. To establish which of the results are significant, we performed the t-test on $\Delta NLLs$ followed by the FDR correction. We highlight in italics the results that did not pass the threshold q-value of $q = 0.05$.

Table 7: Model comparison

| MODEL | $\Delta NLL_{train}$ | $p_{train}$ | $q_{train}$ | $\Delta NLL_{test}$ | $p_{test}$ | $q_{test}$ |
|---|---|---|---|---|---|---|
| 0-THEORY OF MIND | $0.31 \pm 0.06$ | $10^{-6}$ | $10^{-6}$ | $0.44 \pm 0.08$ | $10^{-7}$ | $10^{-6}$ |
| SHARED BELIEFS | $0.56 \pm 0.12$ | $10^{-5}$ | $10^{-5}$ | $0.65 \pm 0.11$ | $10^{-7}$ | $10^{-7}$ |
| FLEXIBLE IN-FIGHT POLICY | $0.46 \pm 0.17$ | $0.01$ | $0.01$ | $0.29 \pm 0.13$ | $0.03$ | $0.04$ |
| RESCORLA-WAGNER | $0.13 \pm 0.12$ | $0.29$ | *0.29* | $0.14 \pm 0.11$ | $0.21$ | *0.21* |
| FIXED PRIOR BELIEFS | $0.57 \pm 0.11$ | $10^{-6}$ | $10^{-6}$ | $0.68 \pm 0.12$ | $10^{-7}$ | $10^{-7}$ |
| FIXED POSTERIOR BELIEFS | $0.52 \pm 0.10$ | $10^{-6}$ | $10^{-6}$ | $0.51 \pm 0.11$ | $10^{-5}$ | $10^{-5}$ |
| SHUFFLED BODY WEIGHTS | $0.26 \pm 0.10$ | $0.02$ | $0.02$ | $0.18 \pm 0.10$ | $0.07$ | *0.08* |
| WIDE BODY WEIGHT PRIOR | $0.12 \pm 0.02$ | $10^{-7}$ | $10^{-6}$ | $0.12 \pm 0.02$ | $10^{-6}$ | $10^{-6}$ |

## C.4 Normative theory of social conflict

Lastly, we summarize our behavioral and neural findings to propose a reward-based model of decision-making during social conflict.

1. Mouse decisions in chronic social conflict can be described with a game theory model where agents choose actions to maximize the perceived reward of winning minus the cost of losing, and the probability of winning depends on the animals' strengths;

2. Animals' behavior is consistent with the perceived cost of losing while attacking being larger than the cost of losing while defending and also larger than the reward for winning;

3. Animals' actions and neural activity are consistent with maintaining the first-order Theory of Mind (1-ToM) endowing animals with primary beliefs about their strengths and strengths of their opponents, and secondary beliefs about the beliefs of their opponents;

4. Initial beliefs, best describing the animals' actions in the model, are based on the body weight for oneself and reflect the population average for opponents;

5. Belief updates, consistent with animals' behavior, are described with Bayes' rule based on observed actions and interaction outcomes;

6. Animals' actions are likely decided before each agonistic interaction and do not adjust to opponents' actions during an agonistic interaction.

These conclusions suggest that pathological aggression and social defeat may emerge from beliefs about own strength, updated over interactions.

