# OpenReview forum: "A normative theory of social conflict"
_NeurIPS.cc/2023/Conference — NeurIPS 2023 poster_

### Official Review · Reviewer_uQtV · 2023-06-28

**Soundness:** 2 fair
**Presentation:** 3 good
**Contribution:** 2 fair
**Rating:** 4
**Confidence:** 3

**Summary:**

The paper presents experiments conducted with real mice, that were put in varying conditions of social stress. In particular, pairs of mice where put together and their behaviour was observed. The attacking mouse was classified as dominant and winner. Initially, loosing mice were move, whereas winning mice stayed. Mice were euthanised and their brains were analysed for brain activity, based on some marker which has been shown to be an indicator.

The results were recoded and various models where fitted to the data in order to explain when mice attacked or defended, based on their believe of the opposing mouse. The data is based on more than 100 mice.

**Strengths:**

The idea of the paper is appealing. Using data of real animal behaviour and try to model it with a game theoretic approach. The sample size is large, compared to other animal studies. The math and results are presented well.

**Weaknesses:**

There are several issues with this paper.

The biggest issue that I have with this paper is that more than 100 mice where ethanised without any discussion of documentation of ethical standards and considerations.

Furthermore, the mice are placed purposely in stressful conditions and it is not clear how the collected data is a good reflection of their beliefs. As an example. Winning mice remain in the environment, while loosing mice are moved to another environment. Yet, attacking is considered winning. As a mouse, not moved to another enviroment, defending could be the more natural behaviour if a new animal is placed in the environment. A male lion defends it's place while the youg male attacks to take it's place. The classification of attacking = winning is not well argumented, which questions the underlying assumption.

The discussion of the broader impact is mainly restating the experiemnts and methods. There is no real discussion what can be learned from fitting the models to the data.

**Questions:**

- Are there any documentation of ethical standards, consideration, etc. which are required for mice experiments?
- What are the conclusiosn from fitting the models to the data?
- Please explain the underlying assumption of the data, why is attacking = winning? What effect does moving mice have as compared to remaining? Are the environments different? Is moving more stressful for the mouse that is moved, if so, why?



**Limitations:**

The authors state "A detailed discussion of our results including the strengths and limitations of our approach is provided in Appendix A". I expect the discussion of the limitations in the main paper.

The title of the paper is "A normative theory of social conflict". I don't see various models fitted but I can't see a normative theory that is derived from the experiments.

---

> ### Author Rebuttal · Authors · 2023-08-09
>
> While we appreciate the Reviewer taking possible ethics concerns seriously, we would like to point out that they’ve been (i) addressed in the original text via the ethics statement in the Methods section [line 95, supplement] and via referencing the experimental paradigm in the Introduction [line 23, main text], (ii) not raised by 4 other Reviewers, and (iii) not confirmed as problematic by 2 Ethics reviews based on the very same text. Please see the details below, alongside our responses to other specific questions.
>
> **As the rating and confidence mean that they’re based on the ethics concern, we ask that the Reviewer re-evaluates the paper to reflect the cleared ethics concern.**
>
> **Ethics concern.** Our experiments match or exceed the requirements equivalent to those of the NIH. Before the start of the project, the experiments were approved by a qualified board specialized in animal research ethics. We will name the exact board and the exact guidelines in the non-anonymous version of this paper. Our experimental design is standard for the field of aggression, which uses mouse/rat behavioral experiments with subsequent c-Fos imaging. Based on a paradigm used for over 25 years (e.g. see Kudryavtseva et al, 2000), similar experimental designs have been approved in a large number of experiments by boards in Europe and the US.
>
> **Number of mice.** To track the progression of aggressiveness/defeat and related neural activity, we needed to evaluate the stages of aggressiveness including ‘normal’ aggression (3 days), ‘pathological’ aggression (20 days), the transition between the two (10 days), and the reversal of aggression (22 days) separately for ‘winners’ and ‘losers’. Each experiment has to come with a control group. Mice could not be shared among experiments because the c-Fos imaging can only be performed once per animal.
>
> **For the other details of the experiment and their justification**, including the moving of the mice, please refer to Kudryavtseva et al 2000, cited in our submission.
>
> **The conclusion from fitting the data / the normative theory that’s derived from the experiments.** Thank you for this question. In the draft version of the paper, we had a formal summary of our findings, framed as a theory of social conflict. While we removed it from the text to save space, it may make sense to re-introduce it to the Appendix as a whole. Please find it below:
>
> 1. Mouse decisions recorded in the chronic social conflict paradigm are consistent with a game theory model where agents choose actions to maximize the perceived reward of winning minus the cost of losing, the probability to win depends on the animals’ strengths, and the uncertainty about strengths is parameterized as probabilistic beliefs;
> 2. The perceived reward of winning, consistent with animals’ behavior, is smaller than the cost of losing while attacking but larger than the cost of losing while defending;
> 3. Animals’ actions and neural activity are consistent with maintaining the first-order Theory of Mind (1-ToM) including primary beliefs about the strengths of all involved and secondary beliefs about the beliefs of their opponents;
> 4. Initial beliefs, best describing the animals’ actions in the model, are based on the body weight for oneself and reflect the population average for others;
> 5. Belief updates, consistent with animals’ behavior, are described with the Bayes’ rule based on observed actions and interaction outcomes, progressing at the unit learning rate;
> 6. Animals’ actions are likely decided in advance and do not adjust to opponents' actions.
>
> **For the discussion of these statements,** please refer to the Supplementary Discussion section of the original text.
>
> **For the broader implications of our results,** please see the general response to all Reviewers.
>
> **On attacking and winning.** We do not equate the attacking and the winning. For example, early in the experiment, both mice would attack but the assigned outcome would be a ‘draw’. Overall, we suggest interpreting the ‘attack’, ‘defend’, ‘win’, ‘lose’, and ‘draw’ terms within the game theory in the way we defined it in the original text.
>
> We are happy to answer any further questions and/or to provide further clarifications.

---

> > ### Comment · Reviewer_uQtV · 2023-08-16
> > **Rebuttal**
> >
> > Dear authors,
> >
> > Thank you very much for the detailed rebuttal.
> >
> > As stated to the area chair, I appreciate and respect the decision of the ethics committee and did not ask for another ethics review of the paper.
> >
> > I have updated my rating of the paper.

---

> > > ### Author Response · Authors · 2023-08-16
> > >
> > > Thank you for taking the time to review the rebuttals and update your valuation of the paper.
> > >
> > > We agree that ethical considerations are important and will try to better convey our responsible approach to animal use in the text, following the recommendations that we have received here.
> > >
> > > We are open to discussing any further concerns and addressing any remaining questions throughout the rest of the discussion period.

---

### Official Review · Reviewer_RYok · 2023-07-06

**Soundness:** 3 good
**Presentation:** 3 good
**Contribution:** 2 fair
**Rating:** 5
**Confidence:** 2

**Summary:**

This paper aims at explaining the underlying principles of social conflict with the Theory of Mind modeling. The authors collected the behavioral data of 100+ mice in agnostic contact events, and fit the parameters of the Bayesian inference model. The experiments show that animals' actions are consistent with modeling both the first-level belief of strengths and the second-level belief of the opponents.

**Strengths:**

The model comparison considers a comprehensive suite of contributing factors. The model’s effectiveness is clearly demonstrated by contrasting different control groups. The success regression of the Bayesian model to the mice data shows that both the first and second-level beliefs are employed for decision-making.


**Weaknesses:**

This paper is difficult for me to make a judgment on. The modeling and the ablation studies are sound. However, the Bayesian belief update framework is already well-studied and there is no new algorithm-wise contribution. The model fitting showing both the first and second-level ToM are used in multi-agent decision-making is not new discovery as well. I’m open to discussion with other reviewers and authors about how to posit this paper and its contribution to Neurips.


**Questions:**

see weaknesses

**Limitations:**

see weaknesses

---

> ### Author Rebuttal · Authors · 2023-08-09
>
> Thanks for your time and comments. For the details on the novel contributions of the paper, please see the general response to all Reviewers. We look forward to editing the paper’s Introduction based on the Reviewers’ suggestions and the upcoming discussion, along the lines outlined in our response to all Reviewers.
>
> We are happy to answer any further questions and/or to provide further clarifications.

---

> > ### Comment · Reviewer_RYok · 2023-08-12
> >
> > Thanks for the authors' rebuttal. I believe that this work can serve as a robust exemplar for scrutinizing animals' social interactions through the lens of the Theory of Mind. I'm happy to increase my score.

---

> > > ### Author Response · Authors · 2023-08-15
> > >
> > > Thank you for your response and for your positive valuation of our work

---

### Official Review · Reviewer_UypE · 2023-07-10

**Soundness:** 2 fair
**Presentation:** 2 fair
**Contribution:** 2 fair
**Rating:** 5
**Confidence:** 2

**Summary:**

The paper models social conflict in mice using a game theoretic Bayesian theory-of-mind model. They find that the mice's actions are most consistent with the ToM model where agents (mice) maintain primary beliefs and secondary beliefs about the beliefs of their opponents.

**Strengths:**

The paper models an interesting dataset of mice in social conflict which includes whole-brain neural data. They do many comparisons between different models and ablation studies in their experiments.

**Weaknesses:**

I saw two main weaknesses in the paper. The first is that the paper needs more explanation of the significance of the results. The introduction is combined with the related work, and in particular, the "introduction" part is only one paragraph while there are four paragraphs describing related work. I couldn't understand what the specific contributions of this paper were compared to previous results. I also wasn't sure about the "so what?" of the paper. I think the authors could move the ablation studies in Sec 3.1.1 to a appendix and expand more about the contributions of their results. The description of the model and how it is fit also takes up a lot of space, some of this could probably be moved to an appendix.

The second weakness is in the empirical evaluation of experiments. I'm confused why the models are evaluated by the change in negative log likelihood on the test set. If I'm understanding currectly the actions and outcomes are discrete (either attack/defend or win/lose).. meaning the ground-truth probability of an action/outcome at a given time is either 1 or 0. So, evaluating the models on log likelihood means that models that make more extreme predictions are preferred. Log likelihood is used to fit the model, but in testing, we just care about the predictive accuracy. Why not just evaluate on predictive accuracy? It would be more interpretable too.

The authors evaluate many models, compute many significance tests, and control for FDR because of the multiple corrections. However, they only seem to report the results for the *best* models: "Our comparisons were iterative: they lasted until a model under consideration outperformed all the other models. Below, we report the results for the final round of these comparisons." There aren't enough details in the paper to understand how exactly they controlled for FDR. Did they only control for FDR for the best model comparisons shown in Fig 4? It seems that there may be a _selective inference_ problem going on here. Generally, it makes sense to report a table in the appendix with all hypothesis tests that were run, the resulting p-values, and their corresopnding q-values. I would ask that the authors add this in their revision for clarity.

Since the main result seems to be that the 1-ToM does better than 0-ToM (at least that's what is highlighted in the abstract), why not also test 2-ToM? Is it possible to recover a 0-ToM model using the 1-ToM model class they've defined? In that case, one would expect that 1-ToM fits the data better (but perhaps doesn't generalize as well). If the authors compared to 2-ToM, it would provide stronger evidence that 1-ToM is the correct model (as opposed to simply having lower log likelihood because of having the flexibility to use more parameters.

I may have misunderstood some of the experimental evaluation, and if so, I am willing to change my mind on the above points.

**Questions:**

See above

**Limitations:**

The strengths and limitations are relegated to the appendix. As described in "weaknesses", I don't think there is enough contextualizing for the motivation and takeaways of the paper. I think the authors should move the limitations/strengths to the main text and move some of the ablation studies/method description to the appendix instead.

---

> ### Author Rebuttal · Authors · 2023-08-09
>
> Thanks for your time and comments. As we respond to specific points below, please also see the general response to all Reviewers.
>
> **Significance of the results and contribution of the paper.** Please see the general response to all Reviewers.
>
> **Log-likelihood vs. prediction accuracy.** The standard way to compare models in behavioral research is using formal comparison criteria, such as the Akaike Information Criterion (AIC) or the Bayesian Information Criterion (BIC); both are computed as (a function of) the number of parameters in the model minus the log-likelihood of this model to generate the original data. Most of our models have the same number of parameters; others have a similar number of parameters. As we only evaluate the difference in AICs (or BICs) between the models, the term reflecting the number of parameters cancels out, resulting in the difference in log-likelihood that we present. We thus follow the standard approach in behavior modeling.
>
> **Log-likelihood and the extreme predictions.** While this is true that the log-likelihood favors extreme predictions as long as they are _correct_, the _incorrect_ extreme predictions are heavily penalized. As a result, the log-likelihood cost function overall favors balanced, non-extreme predictions.
>
> **FDR for non-final comparisons.** Thank you for this comment, this is something that we overlooked. In the revised version, we’ll be happy to present the FDR score computed across all final and intermediate results to avoid selection bias. We ran this analysis and the results seem to hold, although we’ll doublecheck it to make sure that we included all interim results in the study.
>
> **2-ToM.** This sounds like a good idea, thanks. Initially, we only compared 0-ToM and 1-ToM because we thought that the result would favor 0-ToM (consistent with the Reviewer's proposal). Time permitting, we will consider doing the 2-ToM analysis (it may take a long time to script it and ensure that all equations are correct; it also takes an exponentially more extended time to run).
>
> **Recovering a 0-ToM model with the 1-ToM model.** If we understood the question correctly, it is not possible to recover a 0-ToM model using the 1-ToM class under our definition. Our 0-ToM model assumes a game-theory-optimal opponent under the same beliefs, whereas the 1-ToM model separately estimates the opponent’s beliefs and assumes that the opponent assumes _our_ action to be game-theory optimal. We agree that this is an important consideration that would be well-placed in the Discussion. We would like to highlight that the 0-ToM and the 1-ToM model have the same number of parameters, so the latter model shouldn’t be overfitting compared to the former model.
>
> We are happy to answer any further questions and/or to provide further clarifications.

---

> > ### Author Response · Authors · 2023-08-15
> >
> > Dear Reviewer, we wanted to ask whether your concerns have been addressed through this and other discussions - just so that the Authors could provide further responses during the week if necessary. Thanks!

---

> > > ### Comment · Reviewer_UypE · 2023-08-16
> > >
> > > Thank you for the clarifying comments, I'm happy to increase my score to a 5.

---

> > > > ### Author Response · Authors · 2023-08-16
> > > >
> > > > Thank you for taking the time to read our paper/rebuttals and for your response. We look forward to incorporating your suggestions, as outlined in the original review, into the updated manuscript.

---

### Official Review · Reviewer_QTqc · 2023-07-24

**Soundness:** 3 good
**Presentation:** 2 fair
**Contribution:** 2 fair
**Rating:** 6
**Confidence:** 1

**Summary:**

This paper uses behavioral and neural data in mice to develop a game-theoretic and Bayesian theory-of-mind based model of social conflict. Mice are assumed to choose whether to attack or defend optimally (the game-theoretic component) based on their beliefs (and beliefs about opponent's beliefs; the BToM component). The authors show that mice behavior is best explained by the BToM model, rather than simpler models (e.g. pure RL over attack/defend actions) or ablated versions of their model. Finally, neural correlates of the model are identified.

**Strengths:**

- The paper is ambitious, covering both a theoretical model and extensive empirical validation. Based on my limited understanding, it proposes and provides evidence for the role of ToM in a fundamental social conflict paradigm. If the general novelty of the approach is in fact high, I agree this could form an interesting basis for future work.
- The model itself is interesting and seems well-justified (although it took a few reads to understand the structure, see weaknesses).
- The science (experimental paradigm, model comparisons and ablations) seems sound. I appreciated the comparisons to, e.g., simpler R-W reinforcement, which justify the ToM-based approach.

**Weaknesses:**

- Presentation. I found the paper difficult to understand: explanations of the experimental paradigm, model, and neural analysis were all compact and somewhat cryptic. Relationship to prior literature-- especially to contextualize the findings, such as neural correlates-- was terse, and the discussion itself was relegated to the appendix. This made it difficult to understand the contribution.
- The discussion of neural findings (both in the main text and appendix) was particularly brief. I'm vaguely aware of neural imaging of ToM elsewhere (e.g. in human gameplay [1,2] or pedagogy [3]) but these connections aren't discussed. Obviously these studies are done in people not mice, but given the authors' stated ambition to be relevant to human behavior, I think it'd be helpful to sketch out how these findings relate to imaging work there.
- This is more of a note to the authors than a weakness of the scientific contribution itself, but I'm not sure NeurIPS is the best venue for this work. There's a lot of ground to cover here  for 9 pages.  Similarly, I'm not sure how relevant the findings will be for the majority of the NeurIPS audience. I was personally unfamiliar with the experimental paradigms and background literature, so I had trouble judging its significance. A neuro journal with a more expansive format and familiar audience might be a better venue.

[1] https://www.pnas.org/doi/10.1073/pnas.0807721106
[2] https://www.pnas.org/doi/10.1073/pnas.0711099105
[3] https://www.pnas.org/doi/abs/10.1073/pnas.2215015120

**Questions:**

- I found Fig. 1D-F very confusing. Why are there gaps in the data, both on the X-axis ("skipped" days) and Y-axis ("skipped" mice)? Substantially more explication would be helpful here.
- Could you clarify the justification for exposing mice to varying schedules of conflict (i.e. 3 / 10 / 20 / 22 days)? It wasn't clear to me how this related back to the theory, or why it was necessary. Was this solely for the purpose of analyzing neural correlates at different stages of conflict? Providing a diagram or a table illustrating the different conditions and analyses would help clarify this.
- Were nuisance variables (e.g. overall win-loss record) regressed out prior to performing the neural analysis? It seems like you might want to control for this.

**Note**: original score `5`, post-rebuttal `6`

**Limitations:**

No concerns. Review board approval was obtained for animal experiments.

---

> ### Author Rebuttal · Authors · 2023-08-09
>
> Thanks for your time and comments. As we respond to specific points below, please also see the general response to all Reviewers.
>
> **The general novelty of the approach and relevance to NeurIPS.** Please see the general response to all Reviewers.
>
> **Presentation.** We are happy to further work on the text with the coauthors and broader colleagues to make the text more understandable. The current unfortunate terseness was imposed by the NeurIPS page limit and numerous considerations we had to discuss regarding our analysis. Specifically, we are happy to extend the Introduction / related work section along the lines discussed in the general response to all Reviewers here to better contextualize our work for the NeurIPS readers.
>
> **Neural imaging of ToM.** We thank the Reviewer for this suggestion and agree that we should discuss more related fMRI literature in the paper. Sadly, the techniques used in fMRI do not exactly apply to our analysis, as they mostly seek the _differences_ caused by a certain experimental condition and/or the _regression_ of the signal, while we focus on the _correlations_ between different levels of c-Fos with different amplitudes of the beliefs. We have a separate project that uses the same data where the fMRI-style analyses would apply, so we appreciate the literature recommendation.
>
> **Figure 1D.** We would appreciate comments on how to improve this figure! Below, please find our initial thoughts:
> - the three panels were meant to highlight the three phases of aggressiveness: ‘normal’, the formation of ‘pathological’, and the ‘reversal’ (from ‘winner’ to ‘loser’ and the other way around);
>
> - what looks like the ‘skipped days’ meant to separate the data of different days. Additionally, different mice had different numbers of interactions on the same day (e.g., known ‘aggressors’ would be matched with ‘undecided’ mice to facilitate them forming a ‘winner’ or a ‘loser’ phenotype);
>
> - the ‘skipped mice’ on the first three days are actually (mostly) a plotting error, thanks for catching that one! We will fix that in the revised figure.
>
> **Varying participation of mice.** The Reviewer is correct in suggesting that the varying participation of mice was done to enable brain imaging on different stages of conflict: ‘normal’, ‘pathological’, and ‘reversal’). Please find additional details in Supplementary Table 1.
>
> **Nuisance variables.** Unlike in fMRI, neural correlates of these variables are something that is directly looked at in conventional studies of aggression. We did not regress the nuisance variables out in the presented analysis as what _we_ looked for were the _correlates_ of _individual_ model variables and not the brain regions whose activity could be regressed to the _set_ of the model variables. Instead, to make sure that nuisance variables do not affect our results, we looked at the voxels whose activity is directly correlated with various task-related variables. We established that a substantial number of voxels, correlated with 1-ToM beliefs, are not explained by any other considered variables. This design choice was loosely inspired by the body of literature (circa 2000-2015) on the effects of normalizing the fMRI signal, which hasn’t converged to a universally accepted way of processing data, which has convinced us to present what we saw as the most direct and conservative analysis. With that said, before submission, we performed additional analyses where we _normalized_ the c-Fos signal by some nuisance variables (data not shown); the result was similar to the one we presented.
>
> We are happy to answer any further questions and/or to provide further clarifications.

---

> > ### Comment · Reviewer_QTqc · 2023-08-14
> >
> > Thanks for the well-thought out response w.r.t. significance of this work!
> >
> > > Presentation. We are happy to further work on the text... The current unfortunate terseness was imposed by the NeurIPS page limit and numerous considerations we had to discuss regarding our analysis. Specifically, we are happy to extend the Introduction / related work section along the lines discussed in the general response to all Reviewers here to better contextualize our work for the NeurIPS readers.
> >
> > I would strongly recommend using the extra page to do this (and moving methods details to the appendix if needed to give it a full treatment). Better to hook readers early so they're willing to go searching for details, than drown them in the details and make them go searching for the significance :)
> >
> > > Figure 1D-F
> >
> > I would suggest (1) dropping multi-interaction mice (``known aggressor``) mice from the x-axis, so only their first interaction is shown. Assuming their phenotype is stable for the second interaction, this seems like a detail that can be relegated to the appendix. The plot can be re-framed as ``Observed mice phenotypes / day'' rather than showing the atomic interactions. (2) I would then drop the empty columns so that the days abut directly. I think this will make following the visual trends easier, since they'll be continuous instead of interrupted.
> >
> > > Varying participation of mice
> >
> > I would consider not mentioning the full schedule until much later then, e.g. when the c-Fos analysis is discussed. At that point you can say ``additional mice were exposed to truncated schedules to facilitate analysis at different stages'' or something like that.
> >
> > Thanks also for the clarification between c-Fos and fMRI. Not something I'm knowledgeable on.
> >
> >
> > I'm happy to raise my score to a `Weak Accept - 6`. This reflects my general faith that the science is sound (with the unfortunate caveat that most of it is outside my area of expertise). Will leave it to the ACs to judge relevance for NeurIPS as a venue.

---

> > > ### Author Response · Authors · 2023-08-15
> > >
> > > Thank you for your detailed response and for the recommendations on the improvement of the text and the figure. We look forward to incorporating these into the revised version of the paper.

---

### Official Review · Reviewer_hLmb · 2023-07-27

**Soundness:** 3 good
**Presentation:** 3 good
**Contribution:** 3 good
**Rating:** 6
**Confidence:** 4

**Summary:**

This work provides a normative framework for reasoning about the strength and, ultimately, the chance of winning in a social conflict in mice. The analysis supporting the practicality of the framework includes both behavioral and neural data.

**Strengths:**

The topic of the paper, i.e., using AI models for the analysis of behavioral and neural data in animals during social interactions, is really interesting. It also covers a variety of topics and models relevant to the work. Furthermore, it used different types of data (behavioral and neural) in the analysis.

**Weaknesses:**

My main concern about the paper is their claim about the existence of Theory of Mind (ToM) in mice. This is highly controversial in psychology, even for non-human primates, and such a claim needs a lot of control experiments.  Moreover, reasoning about strength is different from mind. In addition, I am not fully convinced that level-1 performs significantly better than level-0 as the cost of parameters and deeper reasoning is not considered. Presentation of the results in likelihood, as opposed to sth more human interpretable such as accuracy, makes it hard to evaluate the fits. Even if the results hold, I think the authors should use another term such as joint reasoning about the opponent, or "1-ToM like" framework, with a detailed explanation about the model mimics 1-ToM models in humans and do not necessarily mean the existence of ToM in mice.

**Questions:**

Mostly reflection of mentioned limitations, but to be more precise:
1- Are there any more human interpretable measures that you could use for your comparisons?
2- Sorry if I missed it, but it looks like that the models are not penalized for number of parameters or depth of computaions? Could you present your results considering these penalties?
3- What is the performance of the model that assumes a shared model between opponents? The authors mentioned that they don't have to be, but did they compare with the fixed version (+ considering it has less parameters)? I think if this shared model works it is still valuable in terms of social models (something like joint attention), and also more convincing to be existed in mice than 1-ToM (and maybe 1-Tom do exists, but as I mentioned above it is very controversial)

Update after rebuttal: from 5 to 6 as mentioned in the comments

**Limitations:**

yes

---

> ### Author Rebuttal · Authors · 2023-08-09
>
> Thanks for your time and comments. As we respond to specific points below, please also see the general response to all Reviewers.
>
> **Existence of ToMs in mice.** We agree that the _existence_ of ToMs in mice is controversial and, therefore, we limit our claims in the text to mouse behavior being _consistent_ with running ToMs, which is a lesser claim. We reflect it in the paper’s title, throughout the text, and in the Discussion. We’ll be happy to stress this distinction more.
>
> **0-ToM vs 1-ToM.** The exciting thing about the game-theoretic model that we used is that the 0-ToM and 1-ToM models have the same number of parameters, so they can be compared directly using formal model comparison methods. As the behavioral data is noisy, we also considered the neural correlates of hidden variables in 0-ToM and 1-ToM. We found unique correlates of 1-ToM in the brain that are not explained by 0-ToM or other task-related variables; this result is consistent with the 1-ToM model. Notably, we also registered a (smaller) number of the unique 0-ToM correlates in the brain, so we do not refute that model either.
>
> **Log-likelihood vs. prediction accuracy.** The standard way to compare models in behavioral research is using formal comparison criteria, such as the Akaike Information Criterion (AIC) or the Bayesian Information Criterion (BIC); both are computed as a (function of) the number of parameters in the model minus the log-likelihood of this model to generate the original data. Most of our models have the same number of parameters. As we only evaluate the difference of AICs (or BICs) between the models, the term reflecting the number of parameters cancels out, resulting in the difference in log-likelihood that we present. We thus follow the standard approach in behavior modeling.
>
> **Human-interpretable measures.** In principle, we could use the prediction accuracy as another readout for models’ performance, although we don’t think that it would be informative. The AIC/BIC, represented by the log-likelihood in our case, is a standard metric for behavior models and is also more sensitive to the models’ performance compared to (binarized) classification accuracy. Therefore, here we present the standard AIC/log-likelihood metric.
>
> **Penalty for the number of parameters.** Most of our models (counterintuitively) have the same number of parameters, so the penalty is not needed (e.g. 0-ToM vs. 1-ToM). The ablation models have 1 less parameter each, and the Rescorla-Wagner has three less parameters. Here we describe why we chose not to put this distinction into a penalty. Consider the AIC criterion: AIC = 2(k – log L) where k is the number of the parameters. Delta k between the models is no larger than 3 (between RW and ToMs), which has to be divided by ~60 mice in a dataset (since we fit the parameters jointly for all mice but evaluate each mouse separately to enable the t-test on Delta AIC). Thus, in our plot, the correction for the number of parameters would be smaller than 3/60 = 0.05, which is negligible compared to the typical Delta log-likelihood. As for the penalty for the model complexity, we’ll be happy to hear the Reviewer’s suggestions.
>
> **Shared model / fixed model.** We are not sure what the Reviewer means by the shared model but, when it comes to the ‘fixed’ model, we performed this analysis (see the ‘fixed prior beliefs’ analysis). It has one less parameter than the ToM models (the AIC correction for this penalty would be 1/60) so, even accounting for the number of parameters, it is still outperformed by the ToM model.
>
> We are happy to answer any further questions and/or to provide further clarifications.

---

> > ### Author Response · Authors · 2023-08-21
> >
> > Dear Reviewer hLmb,
> >
> > we wanted to touch base regarding our response. We hope that it addresses the points raised in your review but, should additional discussion be needed, we are happy to engage in the remaining time.

---

> > ### Comment · Reviewer_hLmb · 2023-08-21
> > **Thanks!**
> >
> > Thanks for your response! By shared model, I meant both agents used the same set of parameters, e.g. when A and B are in the conflict there is one value sd_a to estimate the weight of A used by both, and one sd_b to estimate weight for B where again both used by both. They also both update these parameters.
> >
> > I am happy to increase my score by 1 (make it 6).

---

> > > ### Author Response · Authors · 2023-08-21
> > >
> > > Thank you for taking the time to respond to our rebuttal and for your clarification.
> > >
> > > To address your request, we just fitted the 'shared' model as outlined in your clarification. This modification was easy to implement as, under our model, the parameters are already shared between animals (a total of 7 fitting parameters) but the beliefs were individual for each animal. Here, we further constrained $\sigma_{own} = \sigma_{opponent}$, as proposed by the Reviewer, to ensure that, in each pair of mice, the beliefs about the strength are symmetric.
> > >
> > > We compared the fitted 'shared' model to the 1-ToM model and found that the 1-ToM model explains our data significantly better (testing set: $q = 2 \times 10^{-10}$; training set: $q = 1 \times 10^{-5}$). In hindsight, this result is expected, as equating the standard deviations in the model can be viewed as ablation of our 0-ToM model, further constraining its ability to explain the data.
> > >
> > > Overall, we agree that this is a meaningful comparison as symmetric beliefs in competing agents naturally lead to a straightforward definition of the Nash equilibrium. We would appreciate it if the Reviewer could suggest us a literature reference to include in the paper to further support the conduction of this particular test.
> > >
> > > Also thanks for increasing the score. Sadly, the increase hasn't come through in the main review (maybe an OpenReview glitch). Could you please update it there?

---

### Author Rebuttal · Authors · 2023-08-09

We thank the Reviewers for their time and comments. To our understanding, the main request put forward by the Reviewers is to expand the paper’s introduction, describing the state of the art in aggression/conflict research and specifying our contribution to it. We address this request here in our general response and respond to other questions individually for each Reviewer.

Research of conflict is an established field in cognitive science and neuroscience, asking what processes lead individuals to become pathologically aggressive or pathologically submissive. Among numerous other researchers, we find this question to be of high societal importance, as aggressive/submissive behaviors emerge in various social settings, notably including bullying in schools and hostility in prisons. Understanding the neural and behavioral basis of conflict should ultimately allow us as a society to mitigate the negative consequences of aggression and prevent cases of hostility.

The field of aggression/conflict studies has traditionally been qualitative/experimental. The established approach in the field is to ask a binary question (e.g., whether a certain environmental factor increases aggressiveness, or whether a certain brain region changes its activity upon social defeat). To ask these questions, researchers conventionally observe rat/mouse behaviors and record from their brains. Notably, rodents exhibit aggressive behaviors naturally when co-housed in the same cage (which is standard in animal research), quickly forming hierarchies, so no additional actions/stress is put on animals during these experiments. The state-of-the-art results in the field comprise the lists of environmental factors facilitating aggression and the lists of brain regions whose activity is changed as a result of aggression.

While asking binary questions has revealed many facts about aggressive behaviors, this approach doesn’t tell us _how_ aggressiveness develops in an individual, limiting our ability to predict and mitigate harmful behaviors. Here, we set to address this question using behavioral data modeling (which, to the best of our knowledge, hasn’t been applied to aggressive behaviors) for which we needed expressive enough data (as a result, we had to collect it). Compared to prior research in the field, we:

- collected data that covers the _progression_ of aggressiveness/defeat in individual mice;

- proposed a family of models to describe the learning rule, decision rule, and cost function in aggression;

- fitted the family of models to the data on _dynamics_ of aggression to find a model matching the data;

- based on model fits, described the decision-making related to the aggression/submissiveness;

- analyzed the neural correlates of explicit and _hidden_ variables across the _entire_ brain;

- provided validation for our hidden-dynamics model, relating brain regions to _individual_ variables.

To the best of our knowledge, none of the above has been done in aggression literature. Why is it important? A model of this kind can be extremely useful in providing a clear path to addressing the major question of aggression research, that is, how does pathological aggression/defeat develop and what could be the steps to mitigate it. Straightforward follow-up research, enabled by our approach and results, includes:

- testing the model on newly available data (see Discussion);

- identifying correlates of all aggression-related variables (observed and hidden) in the brain and tracing the connectivity of corresponding brain regions. This may reveal the decision-making neural circuit for aggression, highlight the roles of individual brain regions, and enable the shift from a normative to a circuit model of conflict;

- either model then may be used in modeling social interactions to design policies reducing overall aggression. The result may be useful for the school system to reduce bullying.

While the future directions (i.e. the result of our research) drew attention in neuroscience, the machine-learning framework enabling it (i.e. the way these results were obtained) was among the secondary considerations for this experiment-heavy field. Nonetheless, it’s this novel combination of standard machine-learning tools that has enabled these results and that can be highly useful for modeling other social interactions. We find NeurIPS most appropriate for this goal for the following reasons:

- NeurIPS conventionally encompasses neuroscience, machine learning, and their useful interaction;

- we are aware of a relatively large community within NeurIPS who would be interested in these results (perhaps, the fact that we are aware of them, has excluded them from our Reviewers);

- the conference attendees may benefit from our work being exposed to them as, apart from neuroscience, our novel approach combining game theory and Bayesian beliefs can be useful in designing human-robot interactions, e.g. assisted manipulation or autonomous driving, and recommender systems, e.g. informing a person of the robot’s/service’s capabilities.

Finally, our work belongs to the field of NeuroAI, one of whose aims is to use machine-learning approaches to enable better data analysis in neuroscience. While there is a huge potential for addressing questions of societal importance using ML (as we hope is exemplified by this paper), the NeuroAI approach, being on the interface of the two fields, is somewhat alien to either of them. The only way we can showcase the usefulness of this approach is through representation and we are confident that NeurIPS is the best venue for such representation, featuring the growing NeuroAI community and participants from adjacent fields. Delivering long-awaited results regarding the brain-wide involvement in conflict that would not be possible without machine learning, our paper fits the NeurIPS agenda.

We are happy to participate in the discussion and to answer any further questions.

Thanks for your consideration.

---

> ### Author Response · Authors · 2023-08-16
> **Relevance to NeurIPS**
>
> Dear Reviewers and ACs,
>
> to further frame the relevance of our work to the NeurIPS community, below we list some great examples of NeurIPS papers on similar/related topics over the years, that were accepted to the conference.
>
> - Sequential effects: superstition or rational behavior? **NeurIPS 2008.**
>
> - How prior probability influences decision making: a unifying probabilistic model. **NeurIPS 2012.**
>
> - A Bayesian framework for modeling confidence in perceptual decision making. **NeurIPS 2015.**
>
> - A probabilistic model of social decision making based on reward maximization. **NeurIPS 2016.**
>
> - Deep learning for predicting human strategic behavior. **NeurIPS 2016.**
>
> - Why so gloomy? A Bayesian explanation of human pessimism bias in the multi-armed bandit task. **NeurIPS 2018.**
>
> - Demystifying excessively volatile human learning: a Bayesian persistent prior and a neural approximation. **NeurIPS 2018.**
>
> - A Bayesian theory of conformity in collective decision making. **NeurIPS 2019.**
>
> - Inverse rational control with partially observable continuous nonlinear dynamics. **NeurIPS 2020.**
>
> - A sampling-based circuit for optimal decision making. **NeurIPS 2021.**
>
> While similar to our submission, these papers are among the examples of decision modeling with machine learning methods steadily being a NeurIPS topic of interest. We are happy to provide more references from NeurIPS and/or similar conferences if needed.

---

### Decision · Program_Chairs · 2023-09-21

**Decision:**

Accept (poster)

**Comment:**

This paper links experiments with computational modeling and crosses levels of analysis. The results highlight the role of particular brain regions in relation to a computational model of social decision making. Reviewers and myself appreciated the thorough analysis and the way it linked abstract variables of the computational model to specific brain regions, measured with c-Fos. This is exactly as computational neuroscience is meant to work. It is clearly in scope for NeurIPS.